# Exposome-wide patterns predict brain health in aging

Mostafa Mahdipour [1,2] ✉, Somayeh Maleki Balajoo [1,2,7], Federico Raimondo [1,2], Jianxiao Wu [1,2], Eliana Nicolaisen-Sobesky [1,2], Shammi More [1,2,3], Felix Hoffstaedter [1,2], Holger Schwender [4], Masoud Tahmasian [1,2,5], Simon B. Eickhoff [1,2] & Sarah Genon [1,2,6,7] ✉

Promoting brain health is vital for well-being and reducing healthcare burdens. Brain health as measured with the Brain Age Gap (BAG) - the difference between chronological and predicted brain age- relates to many factors. However, a holistic view, integrating the range of factors an individual brain is exposed to, is missing for understanding how the exposome shapes brain health. After computing BAG as an indicator of grey matter (GM) health, we predicted it using machine learning based on 261 exposome variables (spanning biomedical, environmental, lifestyle, socio-affective, and early life domains) in UK Biobank participants. Exposome data can predict GM health with factors pertaining to cardiovascular and bone health, along with alcohol and smoking, nutrition and diabetes showing greater contribution to the prediction. In such domains, life period and duration of exposure appeared crucial. These findings call for early prevention in cardiovascular and metabolic health to promote life-long brain health.

Healthy brain aging is a critical societal challenge, as it underpins motor and cognitive abilities while also contributing to the reduction of neurodegenerative disease incidence. Promoting brain health is therefore essential not only for enhancing individual well-being and quality of life but also for alleviating the growing burden of aging on healthcare systems worldwide. Addressing this issue requires an urgent focus on global strategies to promote individual brain health, as acknowledged by several recent national initiatives around the world (e.g., the European CSA BrainHealth https://www.brainhealth-partnership.eu/, Healthy Brain Initiative Collaborative https://www.alz.org/hbi-collaborative#about or Brain Health Diplomacy[1]).

At the scientific level, several initiatives have also been settled to identify key factors that influence neurocognitive health. One example among the most rigorous and influential initiatives is the Lancet Commission on Dementia, which regularly publishes reports on risk factors, prevention strategies, and interventions for dementia. In its most recent report, the commission highlighted 14 modifiable risk factors—including depression, diabetes, smoking, hypertension, excessive alcohol consumption, air pollution, and visual impairment—that collectively account for an estimated 45% of potentially modifiable dementia risk factors[2]. While this report provides invaluable insights into dementia prevention, these factors are drawn from disparate studies with varying methodologies, making it unclear how they interact with one another, with additional minor risk factors, and with potential protective factors.

To provide a normative, whole brain, indicator of brain health for each individual, many studies have recently capitalized on a Brain Age framework[3]. With this approach, a gap ("Brain Age Gap" BAG) between the apparent age of the brain (or "biological" age) and the true age (or "chronological" age) is used as an estimator of the brain health of any

[1]Institute of Neuroscience and Medicine (INM-7: Brain and Behaviour), Research Centre Jülich, Jülich, Germany. [2]Institute of Systems Neuroscience, Heinrich Heine University Düsseldorf, Düsseldorf, Germany. [3]Department of Bioinformatics, Fraunhofer Institute for Algorithms and Scientific Computing (SCAI), Sankt Augustin, Germany. [4]Mathematical Institute, Heinrich Heine University Düsseldorf, Düsseldorf, Germany. [5]Department of Nuclear Medicine, Faculty of Medicine and University Hospital Cologne, University of Cologne, Cologne, Germany. [6]GIGA-CRC-Human Imaging, University of Liège, Liège, Belgium. [7]These authors contributed equally: Somayeh Maleki Balajoo, Sarah Genon. ✉e-mail: m.mahdipour@fz-juelich.de; s.genon@fz-juelich.de

individual[4]. This indicator is seductive for population studies because it offers a normative estimator, integrating whole-brain structural patterns while minimizing the influence of traditional covariates/confounders. Furthermore, this framework is particularly insightful in aging populations in which a wide diversity of aging paces can be observed, with greater paces being generally associated with brain pathology, such as dementia[5].

Further understanding and promoting neurocognitive health at the individual level in a precision brain health perspective requires a holistic approach, taking into account a myriad of factors and considering how all these factors taken altogether explain neurocognitive health of individuals from the population. A relevant conceptual framework in that context is the *exposome*, which encompasses the cumulative lifelong exposure to environmental, social, and biological factors affecting brain health[6]. While many of the individual factors mentioned above have long been studied in isolation, the exposome concept insists on their co-occurrence, co-dependence, and potential interactions[7]. The contemporary definition of the exposome encompasses the external exposome with factors such as education, socio-economical deprivation, social support, stress, and environmental features (air pollution, area density or greenness, for instance)[6], but also the "internal exposome" with biomedical aspects such as inflammation. We further propose to use the term *expotype* to refer a person's unique exposome profile. This perspective acknowledges that an individual's expotype can combine risk factors for diseases (e.g., smoking since adolescence) with more protective factors (e.g., regular physical activity since childhood) that may interact with, or mitigate the effect of the former factors on brain health.

Using a BAG-based grey matter health estimator, some studies have emphasized that grey matter health is tightly linked to several aspects of body health in aging populations[8–10], others have highlighted relationships with early life factors[11] and socio-affective/mental health-related factors (e.g., long-term depressive symptoms[12]). Given the wide variety of factors that can influence brain health, some studies have applied a phenome-wide association analysis of BAG[13,14]. With such approach, for each and every available non-genetic variable, its association with BAG is tested separately, typically for a linear association. Such approaches hence do not account for redundancy and interaction among different factors altogether forming the expotype of an individual. These previous studies may thus have missed important factors appearing only in interaction with others in shaping brain health in aging. Indeed, while some factors are protective (e.g., nutrition diet[15]), others may accelerate brain aging (such as smoking and alcohol consumption[16,17]), and they often interact with other factors. For example, the detrimental effect of diabetes on brain health can be mitigated by a healthier lifestyle[18]. Accordingly, the grey matter health of an individual should be seen as the outcome of his/her internal phenotype formed by the interaction between different organ systems, but which also relates or interacts in turn with early life factors, socio-affective factors, as well as lifestyle factors. Yet, currently, a holistic view is missing to better understand how individual grey matter health is shaped by the expotype. Capturing the complex interplay between a range of internal and external factors together forming the exposome and relating it to grey matter health requires multivariate approaches and predictive modeling.

In this study, we employed machine learning approaches to examine the extent to which an individual's expotype can predict grey matter health, as reflected by grey matter BAG, in an aging population. We then investigated how the set of 261 distinct exposome variables spanning across biomedical domains (such as blood pressure, diabetes, illness and cancers, hearing loss, hips circumference, arterial stiffness), mental health/socio-affective domain (such as work/job satisfaction, family relationship satisfaction), socioeconomical factors (such as ethnicity, qualification), early life factors (such as birth weights, maternal smoking around birth), adversity in childhood

factors (such as felt loved as a child, sexually molested as a child), but also lifestyle factors (such as nutrition, diet, alcohol intake, smoking, among others) and specific external environmental exposure (such as sun exposure, noisy workplace, population density in home area) contributes to the prediction. This was possible by leveraging the extensive assessment provided for a large population cohort in the UK Biobank project. Such a machine learning framework with additional replication analyses enables us to draw solid conclusions on the factors that contribute to predicting grey matter health in an aging population.

## Results

### Brain age gap indicator of grey matter health
We focused on grey matter health as a key aspect of brain health (see Methods for the rationale). In this aim, we first designed a high-performing Brain Age prediction model from grey matter volume after assessing different algorithms and parcellation schemes (see Supplementary Table 1). To build an estimator that would later be sensitive to deviance from a healthy reference norm, the models were trained and explicitly tested in a subset of 5025 healthy participants from the UK Biobank (age range 46–82 years, mean 62.12 ± 7.16 years, 2579 females, see Methods). The best model was the Ridge regression algorithm with 1054 grey matter features. When this model was applied to the remaining population data ($n = 34,365$, age range: 44–82 years, mean 63.86 ± 7.57 years, 18,128 females), it demonstrated an overall very good performance in capturing the brain aging process across the UK Biobank whole population with a high correlation between the chronological and predicted age ($r = 0.76$) and a Mean Absolute Error (MAE) of 3.93 years. The gap (BAG) between the predicted and chronological (real) age was then computed to provide an estimator of grey matter health. As could be expected this estimator was normally distributed with a group-averaged mean of zero. In the whole population, 90% of participants had BAG values between −9.20 and +10.18 years. Furthermore, 5% of participants had a BAG value inferior to −9.20 and 5% of participants had a BAG value superior to 10.18. This indicates that while the group-averaged mean BAG was close to zero, some participants showed a positive gap, indicating that their brain was estimated to be older than their chronological age, and some showed a negative gap, reflecting relatively preserved grey matter compared to their chronological age. In other words, a wide range of variations could be observed in grey matter health within the UK Biobank populations. Next, we sought to predict these variations based on the individual expotype.

### Prediction of grey matter health from the expotype
The prediction of grey matter health from the exposome variables was performed in subsets of participants with data available for a wide range of exposome variables. As illustrated in Fig. 1, for our main analysis, we identified a main subset of 3706 participants with data for 261 distinct exposome variables (see Supplementary Table 2, and descriptive tables: Supplementary Table 3 for continuous variables, and Supplementary Table 4 for categorical variables) within the UK Biobank population. We additionally identified a bigger subset of participants for replication who only missed the left and right bone density measurements ("replication subset", 4292 participants, 259 distinct exposome variables; Supplementary Table 2). Third, we also identified a very large subset of participants but with data for only 201 distinct exposome variables ("variables-restricted subset" mainly missing mental health and socio-affective related variables; Supplementary Table 2). These three subsets enable us to examine the stability of our main findings across different subsamples, varying in sample size. The prediction of grey matter health in these subsets was performed using a random forest algorithm, which has the advantage of accounting for non-linear relationships between the exposome variables and grey matter health. However, our results were also

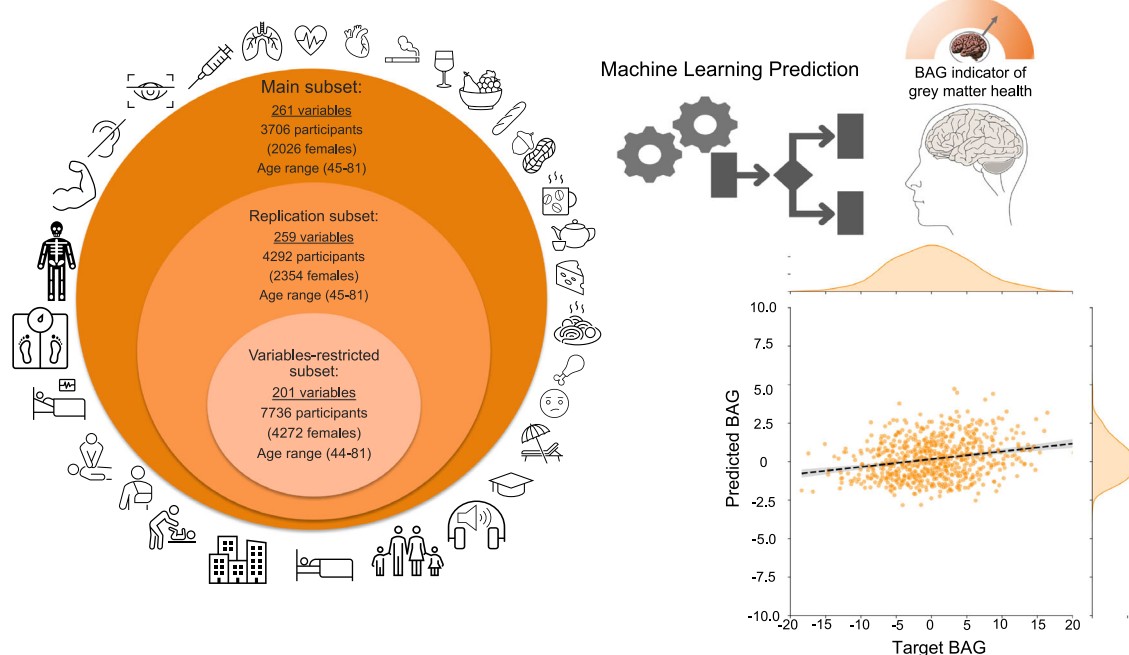

**Fig. 1 | Prediction of the grey matter health (BAG) from the expotype.** Left panel: Machine learning algorithms were trained and tested in three subsets of participants. More than 200 variables were included in each subset, spanning biomedical, lifestyle, socioaffective, early life, and environmental domains. Right panel: grey matter health could be significantly predicted from the expotype. A full list of all exposome variables in the main, replication, and variables-restricted subsets is available in Supplementary Table 2, and detailed prediction performances are available in Supplementary Table 5. In this figure, BAG stands for Brain Age Gap.

replicated using two other popular algorithms: support vector regression (SVR) and ridge regression.

We could significantly predict grey matter health based on the expotype in the main subset ($r = 0.23$; $p = 0.002$; $r^2 = 0.05$; $p = 0.002$, FDR-corrected; Supplementary Table 5). We could replicate this achievement in the replication and variables-restricted subsets (which had less exposome variables but a higher number of participants) with similar accuracies (replication subset: $r = 0.23$; $p = 0.004$; $r^2 = 0.05$; $p = 0.004$; and variables-restricted subset: $r = 0.24$; $p = 0.002$; $r^2 = 0.06$; $p = 0.002$, FDR-corrected; Supplementary Table 5) confirming the robustness of our results across subsamples. Furthermore, grey matter health could also be predicted using alternative algorithms, although achieving numerically slightly lower prediction accuracy (with $r = 0.17$; $p = 0.002$; $r^2 = 0.03$; $p = 0.002$ for Ridge Regression and $r = 0.16$; $p = 0.002$; $r^2 = 0.02$; $p = 0.002$ for SVR; FDR-corrected; Supplementary Table 5). It is noteworthy that p-values were estimated using permutation testing with 500 permutations. Consequently, the smallest $p$-value that can be obtained under this procedure is approximately 0.002, and the true $p$-values may be smaller if a larger number of permutations were used.

### Exposome factors' contribution to the prediction

In order to get more insight into how different exposome factors contribute to the prediction of grey matter health, we used the SHAP explainer. To examine the stability across subsets and algorithms, we ran the explainer on replication and variables-restricted subsets (in addition to the main subset), as well as for additional algorithms in the main subset (see Supplementary Figs. 1–5 and Supplementary Tables 6, 7). The predictive model makes use of the broad spectrum of variables with no specific variable or set of variables showing a disproportionate contribution to the model (see feature importance ranking in Supplementary Tables 6 and 7). In other words, the prediction does not appear to be disproportionately driven by a few specific variables, but instead, it relies on the combination of information across a wide spectrum of variables. However, when looking at the top 30 variables

(Fig. 2), some exposome factors appear particularly important for predicting grey matter health, and this is consistent across prediction algorithms. This group of factors mainly combines cardiovascular factors (such as variables pertaining to blood pressure, but also to smoking) with nutrition, diet, alcohol, and diabetes. Additionally, bone density and hip circumference also appear in the top contributing variables consistently for the main algorithm and one of the alternative algorithms.

Additional insight into the nature of the association between these most contributing variables and grey matter health can be obtained by examining the distribution of SHAP value in Fig. 3. This figure illustrates how the value (from low to high) of the exposome variable is associated with the BAG prediction (from negative which corresponds to better grey matter health to positive which corresponds to poorer brain health). Additionally, to provide some information about effect size and the relationships between the 30 top exposome features and BAG, we performed a traditional linear regression approach. The effect size of each top exposome variable on BAG (when accounting for confounds) is illustrated by plotting standardized beta for each regression analysis in the supplementary fig. 7. It should be noted here that such model does not account for interaction between exposome features and non-linear relationships with BAG. Accordingly, the effect size of any single exposome variable on BAG can be considered as relatively small, in line with the literature on UK Biobank. Whatever the approach, however, overall, most of the associations follow an expected pattern.

Concretely, for factors that are generally considered as risk factors for brain diseases, higher exposure is associated with poorer brain health (i.e., higher BAG). These include high blood pressure, smoking, and alcohol, as well as diabetes. It should be noted here that for these domains, the variables that show the highest impact on the predicted grey matter health pertain to the duration of the exposure and the life periods in which the exposure happens. Hence, for smoking, duration, age at start and age at stop appear as complementary information instead of redundant variables for the prediction. Longer smoking

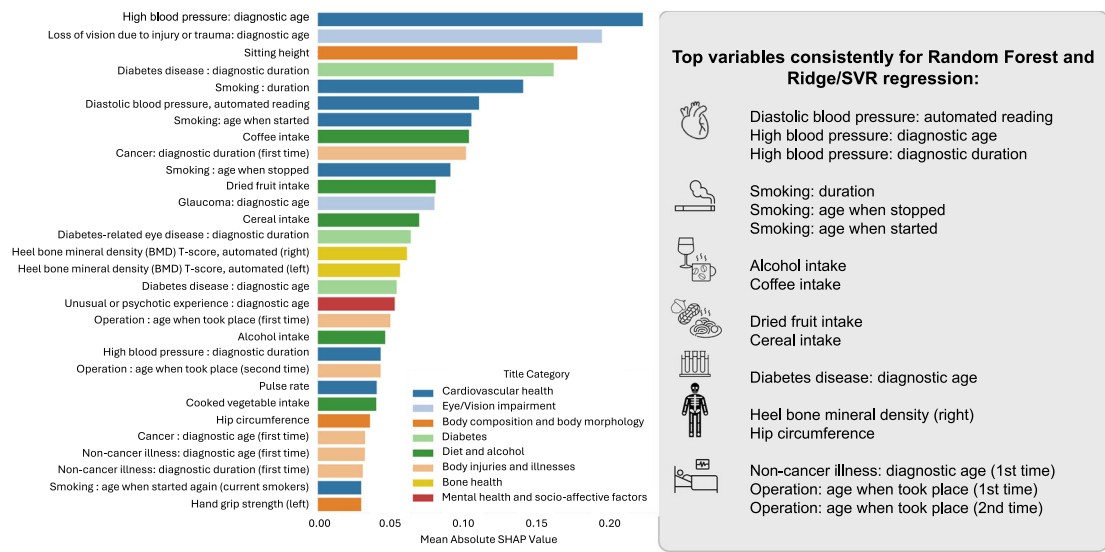

**Fig. 2 | Thirty top contributing variables in the main subset (*n* = 3706, 261 exposome variables).** Left Panel: Exposome variables contribution (as reflected by absolute SHAP value) for prediction of grey matter health based on random forest algorithm. Right panel: summary of the most contributing variables consistently across random forest and at least one other algorithm (see Supplementary Figs. 1 and 2). In this figure, SHAP stands for SHapley Additive exPlanations and SVR stands for Support Vector Regression. Source data are included in Supplementary Tables 6 and 7 and provided as a Source data file.

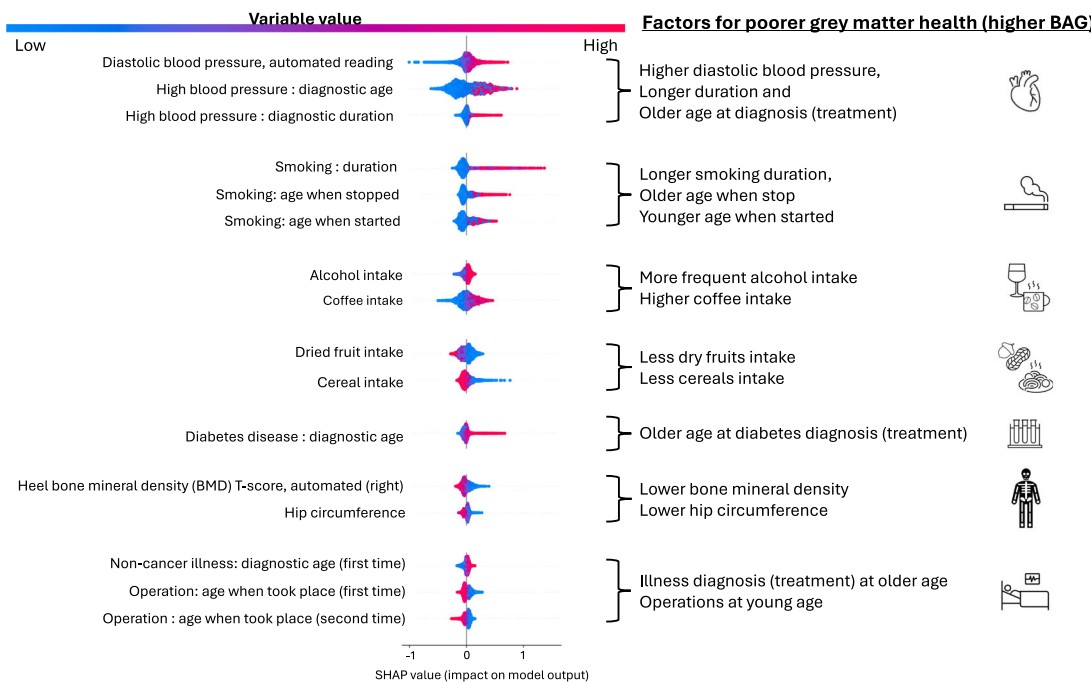

**Fig. 3 | SHAP value for the common most contributing variables when using random forest in the main subset (*n* = 3706, 261 exposome variables).** The *x* axis represents the impact on the prediction: zero could be considered as the default prediction, while a shift to the right reflects a higher predicted BAG (i.e., poorer predicted brain health) and a shift to the left represents a lower predicted BAG (i.e., a better predicted grey matter health). Blue indicates a lower value for the variable, while pink/red indicates a higher value. The right panel summarizes which direction of the variable leads to poorer brain health (i.e., to higher BAG). In this figure, BAG stands for Brain Age Gap.

duration, younger age when started smoking and older age when stopped smoking are all associated with poorer predicted brain health. Similarly, for high blood pressure, a longer duration and an older age at diagnosis (likely reflecting an older age for treatment) both lead to poorer predicted brain health. Along the same line, our results show that when diabetes diagnosis happens at an older age, poorer brain health is predicted. This is also the case for other non-cancer illness: diagnosis (and likely treatment) at an older age is associated with

poorer predicted brain health. In contrast, the relationship between operations (be it first or second operation) and the predicted grey matter health goes in the opposite direction: operations performed at younger age appear to relate to poorer predicted brain health.

For the remaining top contributing variables different patterns can be observed. In particular, for variables related to nutrition diet, while high coffee intake is associated with poorer predicted brain health, low dried fruit and cereal intake leads also to poorer predicted

brain health. Finally, low hip circumference and low bone density are both associated with poorer brain health.

The variables that are lower in the contribution ranking can be found in Supplementary Table 6. It can be noted that some variables do pertain to the general factors listed above, but do not importantly contribute to the model. Examples of such variables are current smoking status (ranked 207 with random forest), vascular/heart problems diagnosed by doctor (ranked 154 with random forest) with the main algorithm, diabetes diagnosed by doctor (ranked 231 with random forest), history of injury caused by alcohol consumption ("Ever been injured or injured someone else through drinking alcohol'", ranked 240 with random forest). In contrast to variables from the same domain appearing in the top contributing group, these variables do not contain information about the life period or duration of the exposure. This pattern suggests that the most important aspect of cardiovascular, metabolic, and lifestyle factors on grey matter health is the life period and chronicity of the exposure.

We can also note that a range of other variables show a relatively low contribution to the prediction of grey matter health. These include mental health related variables (such as history of unusual and psychotic experience, ever had period of mania/excitability and depression-related problems) and socio-affective factors (such as friendships satisfaction), early life factors (such as maternal smoking around birth and "someone to take to physician when needed as a child"), ethnicity, but also specific exposure such as cannabis use and sun exposure. Although these factors can potentially contribute to predicting grey matter health, their influence on the model is relatively negligible.

Finally, these patterns are not importantly influenced by sex. Although in the main analysis, we treated sex as a covariate, in a supplementary analysis, we instead included it as a predictor variable in the model. We observed overall similar results. Furthermore, the contribution (SHAP value) of sex was very low, indicating minimal influence on the predictions. This suggests that the model's performance remains stable and generalizable, regardless of whether sex is accounted for as a covariate or included as a predictor variable.

Taken together, our results suggest that factors pertaining to the internal expotype, including cardiovascular, metabolic, and musculoskeletal aspects, together with smoking, alcohol, and diet, are the strongest predictors of grey matter health. Importantly, variables informing about the chronicity of the life period and the duration of the exposure are the most important for individual prediction.

## Discussion

By considering the exposome reflected in the combination of more than 200 variables spanning different body systems, socio-affective, early life, as well as lifestyle factors, grey matter health can hence be predicted in an aging population. Importantly, the predictive model appears to make use of the wide spectrum of variables. In other words, the contribution to the individual prediction is distributed across variables. This confirms that multifaceted and multivariate views on the exposome are needed to explain interindividual variability in brain health. In that view, the heterogeneity of factors to be considered can be appreciated from our findings on the top contributing variables. These include measurements of the cardiovascular system (pertaining to blood pressure), but also specific metabolic conditions (namely diabetes), common risk factors or body diseases (alcohol and smoking), but also musculoskeletal factors (namely bone density), nutrition diet (namely coffee, dried fruits, and cereals intake), and also medical history of operations/illnesses.

Hence, our results reinforce recent evidence of associations in large aging population cohorts between both the metabolic system and the cardiovascular system, with brain health[8–10,19]. In that context, our study more specifically suggests that alcohol consumption, smoking, high blood pressure and diabetes are the strongest risk factors for altered grey matter. Accordingly, alcohol intake, blood pressure, smoking, and diabetes were strikingly the most consistently reported associations with brain health across previous cohort studies[13,14,20–25]. Our results hence, add to the broad evidence that alcohol consumption negatively impacts brain structure by affecting a range of structural features and being associated with several aging patterns[26–28]. Furthermore, alcohol consumption likely interacts with other risk factors[29] in that context, making it one of the top relevant variables when aiming to promote brain health. Altogether, these results generally point to the fact that the major threats to brain health in our aging populations are metabolic and cardiovascular factors.

However, in contrast to previous phenome-wide association studies of BAG[13,14], our multivariate predictive modeling approach allows us to identify that the top contributing variables for the key other risk factors that are smoking, high blood pressure, and diabetes pertain to the life period at which the exposure has taken place and the duration of the exposure. For smoking and higher blood pressure, we can even note that both, variables pertaining to life period, and variables pertaining to duration conjointly contribute to the prediction. Hence, our multivariate predictive modeling approach allows us to demonstrate that life period and chronicity of the exposure are complementary (rather than redundant) information for determining individual brain health. Overall, the sooner these detrimental exposures are discarded (by stopping smoking and by diagnosing and treating high blood pressure), the lower the negative impact on brain health. Thus, our study crucially highlights the need of early prevention campaigns[30] targeting diabetes, smoking and blood pressure monitoring for public health strategies.

Another important factor from the internal exposome that recently appeared to be associated with brain health in the UK Biobank is the musculoskeletal system[10]. In the current study, we found that bone mineral density, specifically, was often one of the most important contributors for predicting grey matter health. Although the relationship between bone health and brain health has received little attention in the past, evidence for a skeletal-muscle-brain axis exists[31]. For example, low bone mineral density is a risk factor for dementia[32]. In that context, some shared underlying pathophysiological mechanisms have been proposed[33]. In particular, an important neuroprotective role of irisin (a myokine induced during physical exercise by the musculoskeletal system) has been often pointed out as a potential key mechanism in linking bone health to brain health[34]. Thus, the effects on neurocognitive health of multifaceted Interventions (like physical exercise, vitamin, calcium, hormonal therapy, and other medications) that contribute to preserving bone health across aging should be further investigated in future studies.

Beyond the critical effect of body health, our multivariate model also revealed the contribution of nutrition and diet on grey matter health. In particular, coffee, dried fruit, and cereals intake appear in the top most contributing factors consistently across algorithms. A higher coffee intake leads to poorer predicted brain health. Although a neuroprotective role for coffee is often discussed in the literature, our results demonstrate that a daily consumption above the moderate range (around 2 cups/day, the mean of the sample, see summary statistics in Supplementary Table 8) leads to poorer predicted brain health. This finding is consistent with a recent report of positive association between coffee intake and several brain atrophy patterns in the UK Biobank[28]. However, previous phenome-wide association studies did not highlight coffee consumption as strongly associated with BAG when taken in isolation[13,14]. This suggests that the detrimental effect of high coffee consumption on brain health can potentially be explained by interaction with other risk factors discussed above, in particular with cardiovascular factors. It has for example been shown that in UK Biobank participants with genetic predisposition to elevated intraocular pressure, greater caffeine consumption was associated with higher intraocular pressure and higher glaucoma prevalence[35].

Altogether all these results point towards recommending relatively low coffee consumption (not more than two cups/day) for life-long brain health considering that high coffee consumption can detrimentally interact with factors impacting brain health (in particular cardiovascular risk factors) leading ultimately to poorer grey matter health.

In contrast, high cereals and high dried fruit intake generally lead to better predicted brain health. Our findings are aligned with previous evidence that diets in which cereals and dried fruits (such as nuts) are predominant (such as the MIND/DASH diets) have a beneficial effect on brain structure, cognition, and dementia risk[36,37]. It should be noted here that cereal intake mostly reflects whole-grain cereal intake based on the type of cereals that were reported by the participants (see Supplementary Table 9). Several elements (in particular fibers) in cereals and dried fruits are known to have health-improving properties and may even mitigate the effects of diabetes on brain health[38]. Furthermore, dried fruits intake have been demonstrated to play a protective role against cardiovascular disorders in UK Biobank data[39]. Although further studies are needed in the future to better understand how diet and other factors interact with metabolic and cardiovascular factors to influence individual brain health, our current study suggests that high (whole-grain) cereal and dried fruits intake are important factors contributing to promote grey matter health in aging.

In our machine learning models, hip circumference also appears to contribute to the prediction of grey matter health. More concretely, lower (than the mean of 102 cm, see Supplementary Table 10) hip circumference is associated with poorer brain health in the prediction model. However, additional analyses show that, when taken in isolation, hip circumference is not associated with grey matter health in the main subset (null correlation with BAG, see Supplementary Fig. 6). This suggests that hip circumference plays a role as a moderator variable in the individual prediction of grey matter health. In other words, lower hip circumference on its own does not result in poorer brain health, but in interaction with other factors, it can lead to poorer predicted brain health. It can, for example, be assumed that a combination of low hip circumference and low bone density reflects physical frailty, a body condition associated with neurocognitive impairment[40]. In this vein, when hip circumference is taken in conjunction with other variables, it can indicate specifically local adiposity, which can have a protective effect against some metabolic and cardiovascular conditions. In particular, hip circumference is an indicator of gluteofemoral fat which, in contrast to visceral fat, is associated with reduced risks of insulin-resistance, type-II diabetes and cardiovascular problems, as well as, to metabolic health[41,42]. Since we observe here that hips circumference, a proxy for gluteofemoral fat, does contribute to predict grey matter health while it is not directly linearly associated to grey matter health in the sample, we can assume that gluteofemoral fat may modulate the influence of cardiovascular and metabolic risks factors on brain degeneration through complex pathways. This is an interesting question for future studies which should determine exactly how hips circumference/gluteofemoral fat modulates the relationship between cardiovascular-metabolic health and brain structural health.

Finally, some other factors included in the predictive model show relatively null or negligible contributions in the prediction of grey matter health. These factors include mental health-related variables (such as having potentially already experienced unusual and psychotic experiences, depression-related items, and socio-affective factors), early life factors (such as maternal smoking and adversity in childhood), ethnicity, specific exposure such as cannabis use, sun exposure, and home area population density. Although these external factors can be found to relate to grey matter in some studies[43–45], our results show that other factors, in particular those more directly related to body health or the internal exposome, have a greater contribution when it comes to predicting the global grey matter health at the individual level. Thus, when using a multivariate approach that offers a holistic view on the exposome, we can observe that some exposome factors have a relatively minor contribution while others, pertaining to body health and related lifestyle, play a more important role in explaining interindividual variability in grey matter in aging.

In sum, by using a machine learning approach that allows a holistic view, our study has paved the way towards individual prediction of brain phenotype from the expotype. However, several limitations should be noted. First, at the methodological level, in this study we did not account for family relatedness. The UK Biobank includes several participants with first-, second-, and third-degree familial relationships which can potentially bias the results. The genetic relatedness between pairs of participants has been summarized in the pre-calculated kinship coefficients, which could be incorporated in future work to explicitly model or control for familial relatedness. Second, as mentioned in method section, most exposome variables were measured at the initial UK Biobank assessment visit (2006–2010), and when data were not available at that time point, we used measurements from the first repeat assessment visit (2012–2013). We did not explicitly model the time interval between exposome assessment and the imaging visit. To mitigate potential effects of temporal variability, we prioritized measures reflecting relatively chronic exposures (e.g., weekly dietary intake rather than 24-hour recall) and derived duration-based variables where possible (e.g., smoking duration, disease duration). Nevertheless, this approach assumes that modest between-subject variation in the timing of exposome assessment relative to imaging does not substantially influence the observed associations. This assumption cannot be formally tested within the present study and therefore represents a methodological limitation. Future longitudinal analyses explicitly modeling time-varying exposures and exposure–outcome lag effects will be important to further clarify these temporal relationships.

Third, while the prediction accuracy in the out-of-sample test set is significant and demonstrates the model generalizability, the effect size ($r$ value) remains relatively moderate. Relatedly, the explained variance in brain structural health (i.e. in BAG) in the test set is relatively limited (~5%). Although the model generates predictions and feature attributions at the individual level, the current level of predictive accuracy does not support personalized inference or individual-level decision-making. Accordingly, individual SHAP explanations are presented to illustrate model behaviour rather than to enable personalized interpretation. The potential utility of exposome-based machine learning models for individual-level vulnerability or resilience prediction should therefore be considered as a future perspective, contingent on substantial improvements in predictive performance.

Along the same line, at the level of individual exposome variables, classical regression-style effect sizes (e.g., years of brain ageing per unit increase in exposure) cannot be directly inferred from the multivariate machine learning models used here, due to their non-linear structure and the complex interactions between predictors. To facilitate interpretation of the direction and shape of associations, we therefore relied on SHAP-based explanations, which illustrate how changes in exposure values are associated with increases or decreases in predicted BAG across their observed ranges, but do not provide effect sizes in the traditional regression sense. When considering each of the top exposome variables individually in relation to BAG (Supplementary Fig. 7) using a traditional linear regression model, the effect size of any single exposome variable on BAG can be considered as relatively small, in line with the literature on the UK Biobank. Overall, this reinforces the view that brain health emerges from the cumulative and interacting effects of many factors rather than large effects of single exposures.

Nevertheless, the overall low amount of variance explained in a test (unseen) sample suggests that additional predictors should be taken into account. These should include more fine biological measurements (such as those based on biofluids) beyond the basic

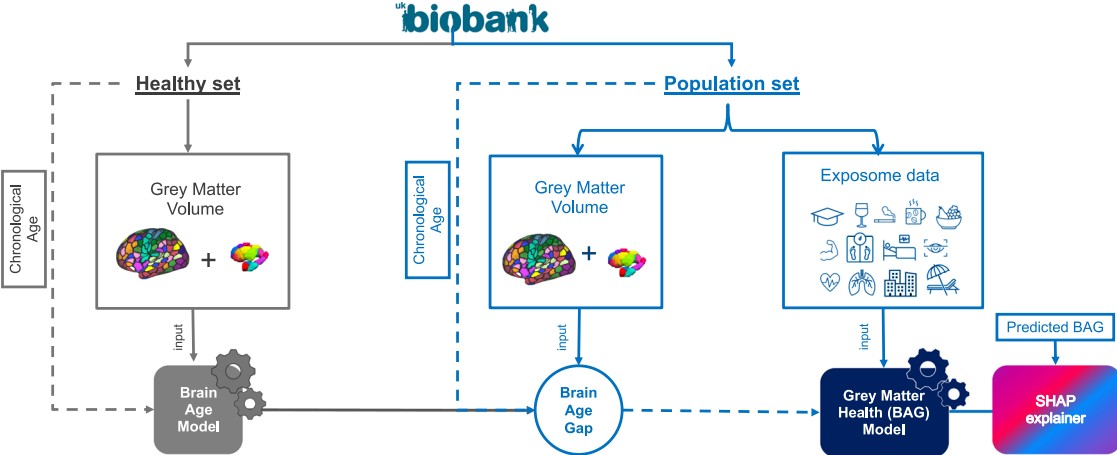

**Fig. 4 | General workflow in the UK Biobank.** Two machine learning models were developed: a Brain Age Prediction Model was developed from a healthy set of UK Biobank participants ($n$ = 5025, 2579 females), while a Grey Matter Health (BAG) Prediction Model based on exposome data was developed in participants who have not been included in the healthy set to avoid any data leakage. Continuous lines indicate features/predictors fed into the model, while dashed lines are used for target variables. Three subsets were further created from the population set to evaluate grey matter health (i.e., BAG) prediction models. The UK Biobank logo is used with permission. Cortical and subcortical atlas visualizations are based on the Schaefer cortical atlas (open-source license) and the Melbourne subcortex atlas, both cited in the Methods.

biological information on which we focused in this study for the sake of optimizing the sample size. In the same vein, our study points to additional factors (coffee intake, dried fruits intake and hip circumference) that seem to interact with metabolic and cardiovascular risk factors in predicting brain structural health. However, although we initially considered body and lifestyle factors from the lens of the exposome, many factors (such as metabolic and cardiovascular conditions) may obviously be importantly influenced by genetic factors. Future studies should thus crucially also include genetic factors, which have been found to also partly explain interindividual variability in brain health measured with BAG[14,46,47]. Leveraging genetic profiles to examine to which extent the combination of the individual genotype and the individual expotype can predict individual brain health in aging with predictive modeling would hence be a crucial step towards precision brain health.

It should also be acknowledged that although BAG is a very convenient indicator of the degree of relative preservation or degeneration of a whole grey matter pattern, it remains a derived indicator instead of a direct measurement, which implies some noise. Furthermore, its conceptual validity and interpretability remain importantly discussed[48]. First, a BAG framework assumes a unique aging trajectory and deviance (older brain age) from this trajectory is considered as advanced or pathological aging[48]. Yet, this assumption can be questioned. Second and relatedly, some studies suggest that deviance (older brain age) may reflect brain phenotype present since early life instead of an aging phenotype[49]. Overall, these considerations question the extent to which BAG can be used as an indicator of brain health in aging. With these conceptual limitations in mind, in this study, we did not use the popular "brain (biological) aging" concept when referring to BAG on purpose because it is not clear whether BAG represents any single phenomenon or set of phenomena that can be referred to as aging[48]. Actually, more research on the underlying neurobiological properties detected by brain age prediction will be required to differentiate which phenomena BAG is in fact sensitive to. In that perspective, systematic comparison with other markers of neurocognitive health in future studies could contribute to better understand the neurobiological phenomenon captured by each marker and how each and every marker is shaped by the exposome.

Despite these limitations, the current study revealed a set of factors primarily contributing to predict structural brain health in an aging population, highlighting cardiovascular health, bone health, and diabetes as key factors, confirming previous studies based on univariate, linear, and not cross-validated modeling[13,14,20]. More importantly, our study pointed to additional factors, which may not show strong linear association to brain structural health when taking in isolation and which therefore are assumed to interact with other factors, such as nutrition diet and coffee intake, as well as hip circumference. Overall, this study set a primary framework for precision brain health and public health policies by revealing the spectrum of mostly modifiable factors explaining variability in grey matter health. In the future, inclusion of genetic factors should enable more precise prediction and a better understanding of how these factors interact with exposome factors in defining individual brain structural health. Such progress would help to better clarify individuals at risk. In this perspective, including more diverse cohorts with regard to ethnicity and geographic areas should also be considered. This would contribute in the future to a better understanding of how sociodemographic factors interact with others in explaining interindividual variability in brain health[6,50].

## Methods

### General workflow

Machine learning models were developed in two main steps in this study. First, a brain age prediction model based on grey matter data was developed. This model was needed to derive a grey matter health indicator (BAG) based on the discrepancy between the predicted and the chronological age. Second, a grey matter health (BAG) prediction model was developed based on exposome data. Both steps were carried on in the UK Biobank, but the two models were developed in strictly separated subsets of participants (to avoid data leakage) as illustrated in Fig. 4 below.

To develop a brain age prediction model, we identified a large subset of 5025 (cognitively) healthy participants ("healthy sample") within UK Biobank. Within the remaining population of the UK Biobank, we identified one main subset, a replication subset and a "variables-restricted" subset of participants (see Supplementary Table 2) to which the previously developed brain age model was applied to compute a BAG for each and every participant. These subsets were defined based on the availability of exposome variables in the participants (see Fig. 1). In the second step, machine learning models were developed to predict grey matter health (i.e., BAG) in the participants of these three subsets.

## Participants

Detailed explanation of data collection in UK Biobank cohort can be found in https://www.ukbiobank.ac.uk/media/gnkeyh2q/study-rationale.pdf. This study focuses on participants with available imaging data (39,390 participants, aged 44–82 years, mean 63.64 ± 7.54 years, n = 20,707 females). Written informed consent was obtained from all participants. The present analyses were conducted under data application number 41655 and were approved by the ethical committee of the Heinrich Heine University Düsseldorf.

Cognitively healthy participants ("healthy sample") had no self-reported long-standing illness disability or infirmity (UK Biobank data field #2188), no self-reported diabetes (field #2443), no stroke history (field #4056), no ICD-10 diagnosis and good or excellent self-reported health (field #2178). These criteria were defined in line with a previous study[20] and led to a sample of 5025 healthy participants (age range 46–82 years, mean 62.12 years ± std 7.16 years, 2579 females), while leaving 34365 participants (age range: 44–82 years, mean 63.86 ± 7.57 years, 18,128 females) to define subsets for exposome based prediction of grey matter health (see Supplementary Table 2).

## Grey matter features

To obtain an estimator of grey matter health, we used grey matter volume features. It should be noted that when using a Brain Age Gap Indicator, depending on the neuroimaging features which are used, different neurobiological aspects of brain health are probed. Many recent studies in large neuroimaging databases have capitalized on multimodal neuroimaging data (typically including functional, white matter, and grey matter features) to derive a Brain Age Gap estimator. This combination of different features was generally for the sake of optimizing the model's accuracy[4], but it results in a global brain health indicator (combining different aspects of brain structure and function) that may lack specificity from a neurobiological standpoint. In contrast, unimodal estimators, such as those explicitly based on grey matter, can be derived with close accuracies to indicator derived a multimodal model[4,20] while providing a more specific insight into the healthiness of grey matter and its relationship to different factors.

Comprehensive information regarding the neuroimaging data from the UK Biobank is available here: https://biobank.ctsu.ox.ac.uk/crystal/crystal/docs/brain_mri.pdf. T1-weighted MRI images were acquired by 3 Tesla Siemens Skyra scanners with 32 channel head coils and used an MPRAGE sequence with 1-mm isotropic resolution, with Field-of-view: 208´256´256. For Imaging data preprocessing, we used an in-house developed framework for computationally reproducible processing of large-scale data (FAIRly big[51]). For this purpose, a singularity container with a pipeline to perform voxel-based morphometry (VBM)[52] on individual T1-weighted MRI images based on the Computational Anatomy Toolbox (CAT)[53] was created. All T1-weighted anatomical images were processed with the CAT version 12.7. After normalisation and segmentation, the grey matter volume segments were modulated for non-linear transformations and smoothed. Grey matter was parcellated using a combination of the Schaefer atlas for 200, 400, 600, 800, and 1000 cortical regions[54] and the Melbourne subcortex atlas for 32 and 54 subcortical regions[55] leading to five levels of representations of grey matter (grey matter volume for either 232, 454, 654, 854, and 1054 regions). After evaluation of brain age prediction performance with grey matter features at these different levels of granularity (see Supplementary Table 1), the atlas of 1054 regions was selected as the optimal grey matter representation for BAG computation in the UK Biobank population.

## Brain age prediction model

Four predictive algorithms were evaluated to design a Brain Age Prediction Model. These included Linear Regression, Ridge Regression[56], Support Vector Regression (SVR)[57,58], and Random Forest (RF)[59] as implemented in scikit-learn[60] and Julearn packages[61]. They were all trained to predict an individual's chronological age using the five different sets of grey matter features (with different levels of granularity) in healthy sample (n = 5025). Predictive brain age models were trained on 80% of the healthy sample (n = 4020) using 10-fold nested cross-validation with 5 repeats for estimating the chronological age based on the five different sets of individual's grey matter features. Furthermore, a 10-fold inner cross-validation loop was implemented for hyperparameter tuning using a grid search approach. The optimized brain age models were then validated on held-out set made by the remaining 20% of the healthy sample (n = 1005) (Supplementary Fig. 8A). Model performance was quantified using the Pearson correlation coefficient (r) and mean absolute error (MAE) between predicted and chronological age in the held-out set.

Finally, the model with the minimum prediction error in held-out set was selected and fitted on the entire healthy sample (training + held-out) and used to estimate the chronological age in population set (n = 34,365, age range: 44–82 years, mean 63.86 ± 7.57 years, 18,128 females) in the UK Biobank dataset. According to the results presented in the Supplementary Table 1, the best model was Ridge regression with 1054 grey matter features.

Importantly, several studies have brought attention to an age bias/age dependency in brain age prediction requiring a so-called age-bias correction[4]. This was done here by regressing out the effects of age on the predicted age[62]. Specifically, we calculated a regression line with chronological age as the predictor and predicted brain age as the outcome, using only the training dataset. The resulting slope and intercept were then used to adjust the predicted brain age values in the testing dataset. This adjustment involved subtracting the intercept from each predicted brain age and then dividing by the slope. In other words, the slope and intercept of the regression line between chronological age and the predicted age were estimated in the training set and used to adjust the predicted age in the test and the population samples. As expected, and illustrated in Supplementary Table 11, this age-bias correction procedure substantially reduced the artefactual correlation between the subsequently computed BAG and age.

## Brain age gap (BAG) as an indicator of grey matter health

The predicted age was then used to compute an indicator of grey matter health for each individual participant based on the discrepancy between his/her (true) chronological age and the "apparent" (i.e., predicted) age of his/her brain grey matter. This was done by calculating the so-called "brain age gap" (BAG) by subtracting the actual chronological age from the adjusted predicted brain age. Accordingly, it provided a normalized measure of the extent to which an individual's brain appeared older (BAG > 0) or younger (BAG < 0) than same-aged peers. The BAG offers a distinctive advantage as it is a personalized measure by nature. Moreover, it is cross-validated and does not convey age-related information (or age-related statistical signal) following age-bias correction, allowing it to directly assess deviations from population norms. In agreement with this view, this estimator was normally distributed in the UK Biobank whole population, as well as within our main subset, the replication subset and the variable-restricted subset (see Fig. 5). In other words, across all samples, most participants showed an almost null gap (i.e., an apparent brain that corresponds to their chronological age) while some participants show a positive gap indicating that their brain is estimated older than their chronological age and some other participants show a negative gap hence reflecting relatively preserved grey matter compared to their chronological age.

## Exposome variables

To represent the exposome, we used a wide range of non-imaging variables encompassing biomedical, lifestyle, socio-economical and early life factors as illustrated in Fig. 1 and the full list of exposome variables are available in Supplementary Table 2. The selection of

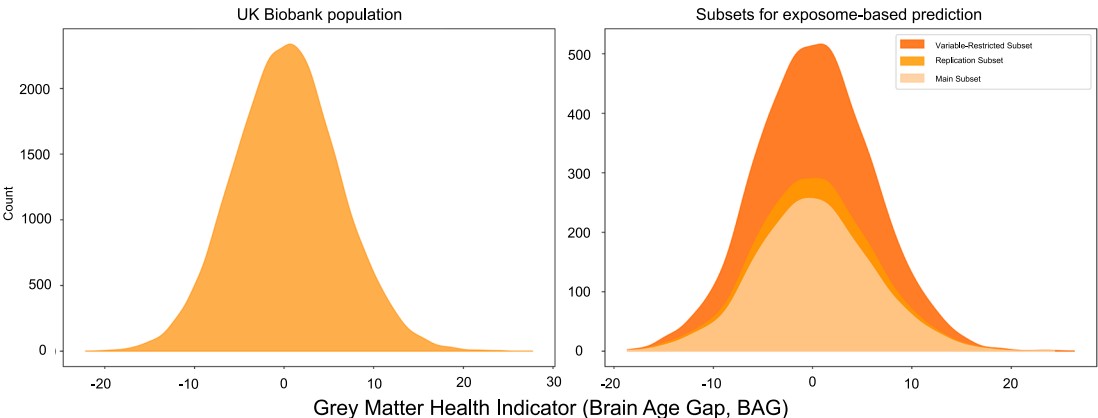

**Fig. 5 | Distribution of the brain age gap across the UK Biobank population and analysis subsets.** Distribution of the brain age gap in the remaining UK Biobank population (i.e. excluding the healthy subset in which the Brain Age prediction model was developed; $n = 34365$, age range: 44–82 years, mean 63.86 ± 7.57 years, 18128 females, left panel) and in the specific subsets of participants (main subset, $n = 3706$; replication subset, $n = 4202$; variable-restricted subset, $n = 7736$, right panel) that were used for exposome-based prediction of grey matter health (see Supplementary Table 2).

variables was guided by previous publications reporting either associations (with correlation or regression usually) with brain health as measured by the brain age gap[10,11,18–21,25,63,64] or associations with grey matter structure[26,65–69] in the UK Biobank. This choice was also constrained by data availability. To avoid potential biases associated with substantial missing values across participants, we focused on variables that were available in at least 2000 participants including both males and females. Of note, we carefully considered multiple imputation. However, the missingness structure in the dataset posed substantial challenges. Specifically, participants with neuroimaging data exhibited block-like patterns of missingness, resulting in limited overlap across many variables (see Supplementary Fig. 9). Under such conditions, chained-equations multiple imputation would be prone to instability and strong model dependence, as many conditional models would need to be estimated from sparse shared information. Moreover, implementing multiple imputation across several hundred heterogeneous variables (continuous and categorical) within a cross-validation framework—while avoiding data leakage—would substantially increase both methodological complexity and computational burden, with uncertain gains in bias reduction. Considering these limitations, we adopted a minimum-availability filtering strategy, prioritizing variables with sufficient observations while maintaining analytical tractability.

We also performed some additional data curation processes, which included: uniqueness constraint (we ensured that each field ID and participant was represented uniquely in the dataset and we removed potential duplicate records), aberrant value exclusion (we verified that all variable values fell within the valid range or allowed set of categories e.g., no out-of-range numerical values or invalid category codes), and uninformative value exclusion (we excluded implausible or non-informative responses such as "do not know" or "prefer not to answer"). Furthermore, we also calculated duration variables for medical conditions by subtracting onset age from the age when attended assessment center (chronological age). Finally, it is noteworthy to mention that most exposome variables were measured at the initial UK Biobank assessment visit (2006–2010), and when data were not available at that time point, we used measurements from the first repeat assessment visit (2012–2013).

As illustrated in Fig. 1, these procedures led us to define a main subset in which 261 distinct exposome variables were available for 3706 participants. By dropping out two variables (left and right heel bone density), we could create a bigger subset of 4292 participants that served as a replication subset. Finally, we also identified a bigger subset of 7736 participants with data for 201 distinct exposome

variables. In this "variables-restricted subset", mainly variables related to socio-affective and mental health domains were missing compared to the main and replication subsets. Standard preprocessing was performed on exposome variables, including variables standardization (within the cross-validation scheme to avoid any data leakage). The redundancy between the 261 exposome variables was relatively low as illustrated by the correlation heatmap (Supplementary Fig. 10). Although some clusters of correlated variables can be observed, these pertain to instrumentally related body morphology and composition variables and overall, a substantial degree of heterogeneity can be observed. The distribution of the variables across different exposome domains is illustrated for the reader's information in Supplementary Fig. 11. Of note, from this set of variables, we did not perform any redundancy reduction since we subsequently used machine learning algorithms which are designed to be able to weight or select relevant features (see next section).

## Grey matter health prediction model

To predict grey matter health at the individual level, we first implemented a random forest algorithm, which used a decision tree-based approach and has the advantage of accounting for non-linear relationships between the exposome variables and grey matter health. Random Forest was trained and tested for each subset (i.e. the main subset, the replication subset and the "variables-restricted" subset separately). However, for the sake of replication, we also implemented two additional popular algorithms: ridge regression (which was also used for the brain age model) and support vector regression (SVR). We selected these three algorithms because they offer complementary strengths: random forest as an ensemble method, has the ability to capture non-linearity and complex interaction among features. Furthermore, by combining multiple decision trees, it usually improves prediction accuracy and robustness. While ridge regression provides regularization to prevent overfitting, SVR is very popular and relatively more robust to outliers than ridge regression.

Each additional algorithm was trained and tested on the main subset. Each model was trained using BAG as the target variable (representing grey matter health) and exposome variables as input features within a nested cross-validated scheme. This ensured unbiased performance estimation, with 5 inner folds for hyperparameter tuning and 5 outer folds with 5 repeats to estimate generalization performance. Grid search and Optuna search[70] was employed in the inner folds for hyperparameter tuning.

To mitigate the disproportionate influence of larger-scale features, standardization was applied using the Standard Scaler function

from scikit-learn package[60]. Controlling for covariates was performed on the exposome variables within the cross-validation framework (using estimated from the train set). The set of covariates/ confounds include age, square of age, sex, height, and volumetric scaling from T1 head image to standard space. Thus, all care was taken to neutralize the influence of usual confounds by using a normative indicator of grey matter health and by additionally controlling for covariates within the prediction analysis. It is worth noting that sex was defined based on UK Biobank data-field 31, which reflects sex recorded at recruitment and may include updates provided by participants. Sex was included as a covariate to control for potential confounding effects, but sex-stratified analyses were not performed, as the aim of the study was to model population-level patterns of brain health rather than sex-specific effects.

Model performances were evaluated using multiple metrics, including the mean absolute error, mean squared error, root mean squared error, $r$-square, and Pearson correlation between predicted BAG and target BAG from the test sets for each outer fold. For each subset and each algorithm, permutation test was performed to assess significance against a null distribution by shuffling the scores of the Grey Matter Health variable in 500 repeats of 5-fold cross-validation. Multiple comparisons across different metrics were corrected using false discovery rate (FDR; Benjamini and Hochberg 1995) of $p < 0.05$. Additionally, after selecting the best model, SHapley Additive exPlanations (SHAP)[71,72] were employed to provide an insight into how individual exposome variables contribute to the prediction (Supplementary Fig. 8B). Indeed, SHAP offers a cohesive framework for interpreting predictions in explainable Artificial Intelligence, assigning importance to each variable contributing to a prediction. The stability of SHAP patterns across folds is demonstrated in Supplementary Fig. 12. All analysis steps were conducted using Python, SHAP[71,72], scikit-learn[60], and the Julearn package[61]. Further information on research design is available in the Nature Portfolio Reporting Summary linked to this article.

### Reporting summary
Further information on research design is available in the Nature Portfolio Reporting Summary linked to this article.

### Data availability
The brain imaging and exposome data used in this study are derived from the UK Biobank resource under application number 41655. These data are available under restricted access due to ethical approval requirements, participant consent, and data protection regulations. Access can be obtained by bona fide researchers through application to the UK Biobank Access Management System (https://www. ukbiobank.ac.uk/), subject to approval by UK Biobank. Raw participant-level data are protected and cannot be publicly shared due to data privacy laws. Processed data that do not contain participant-level information are provided with this paper as Source Data files where applicable. Source data supporting this study are provided with this paper. Source data are provided with this paper.

### Code availability
The analysis code used in this study is publicly available on GitHub at (https://github.com/MostafaMahdipour/Predicting_Brain_Age_Gap_ BAG_using_UKB_exposome) and is archived on Zenodo at (https:// doi.org/10.5281/zenodo.18606659)[73]. The archived version (v1.0.0) corresponds to the code used for the analyses reported in this manuscript.

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

## Acknowledgements

This research has been conducted using data from the UK Biobank (https://www.ukbiobank.ac.uk/), a major biomedical database. We are grateful to UK Biobank for making the data available and to all study participants, who generously donated their time to make this resource possible. This work was supported by the Deutsche Forschungsgemeinschaft (DFG, GE 2835/2–1, GE 2835/9-1, SFB 1451 – Project-ID 431549029 to S.G). S.M.B is supported by the EBRAINS 2.0 Project funded from the European Union's Horizon Europe Programme under the Specific Grant Agreement No. 101147319.

## Author contributions

Study concept and design were by M.M., S.B.E., and S.G. Data preparation and preprocessing were carried out by M.M., E.N.S., F.H., and S.M.B. Model development was by M.M., S.M.B., F.R., and S.G. Data interpretation was by M.M., S.M.B., M.T., S.B.E., and S.G. Drafting of the manuscript was by M.M., S.M.B., and S.G. M.M. conducted the analyses, prepared the tables and figures. S.M., J.W., F.R., S.M.B., and S.G. contributed to the statistical/machine learning analyses. Critical revision of the manuscript for important intellectual content was by M.T. and S.B.E. M.M., S.M.B., F.R, J.W., E.N.S, S.M., F.H., H.S., M.T., S.B.E. and S.G approved the final version of the manuscript.

## Funding

## Competing interests

The authors declare no competing interests.
