## [Transparent Peer Review file · Nature Communications]

Exposome-wide patterns predict brain health in aging

Corresponding Author: Mr Mostafa Mahdipour

Version 0:

Reviewer comments:

Reviewer #1

(Remarks to the Author)

In their study, Mahdipour et al. seek to shed light on how a broad spectrum of biomedical, lifestyle and psychosocial exposures jointly relate to individual differences in “brain age gap” — the discrepancy between an individual's age predicted from structural MRI data and their chronological age. Here, brain-age gap is used as a proxy for overall brain health. The authors implement a two-step pipeline using data from the UK Biobank: (i) three machine learning algorithms (ridge regression, support vector regression, and random forest) are trained on grey-matter parcels from 5,025 healthy adults to predict brain age in 34,365 independent participants; (ii) a subset of these participants, scored on up to 261 exposome variables spanning cardiovascular-metabolic, behavioural, socio-economic, and early-life domains, are used to predict the brain-age gap via a random forest model, with SVR and ridge regression used as sensitivity analyses. Model performance is assessed using nested cross-validation, and feature importances are quantified using SHAP values. The exposome explains approximately 5% of the variance in brain-age gap ($r \approx 0.23$, $p = 0.002$). Key contributors include alcohol intake, cardiometabolic burden (e.g., blood pressure history, age at diabetes onset), smoking behaviour, musculoskeletal factors, and dietary patterns (e.g., coffee consumption), while mental health, socio-affective, and environmental variables contribute comparatively little. The authors conclude that modifiable lifestyle and metabolic exposures modestly—but reliably—affect overall brain health, emphasizing the importance of early prevention in cardiovascular and metabolic domains to promote healthy brain ageing.

The manuscript is well-written, methodologically sound, and the analyses are logically organised. I particularly commend the authors for releasing their full codebase in a public GitHub repository, which greatly enhances transparency and reproducibility. The topic is timely and relevant to researchers in neuroscience, ageing, and epidemiology. A notable strength is the study's explicit attempt to disentangle the individual contributions of a broad range of exposures to brain health, thereby illuminating modifiable risk factors. Nevertheless, the work's capacity for causal inference is limited, and several of the associations—though now examined in a multivariate framework—have been reported previously in similar UK Biobank analyses. That said, the work still adds valuable evidence and provides timely insights into modifiable influences on brain health. In the following, I outline a number of concerns that, if addressed, would further strengthen the manuscript.

[1] Novelty and prior work: A broad spectrum of phenotypic associations between lifestyle, health factors, and brain ageing have already been explored in UK Biobank data (e.g., Smith et al.: <https://doi.org/10.1016/j.neuroimage.2019.06.017>; Jawinski et al.: <https://doi.org/10.1101/2023.12.26.23300533>). How do the current findings extend or differ from these prior studies? Please clarify how the present multivariate approach advances beyond earlier univariate correlation analyses.

[2] Variable selection and interpretation: It remains somewhat unclear how the 261 exposome variables were selected. Were these variables chosen purely based on availability and relevance, or were they also selected to represent independent clusters of health-related domains? How distinct are the included variables from one another in terms of conceptual overlap and statistical collinearity? Providing a hierarchical clustering analysis or correlation heatmap could help demonstrate the structure and independence of the variable set.

[3] Training sample characteristics: In lines 120-125, the age range is reported for the test sample, but not for the training sample. For clarity and consistency, it would be helpful to also provide the age range of the training sample, ideally in the same section.

[4] Prediction accuracy and implications for BAG: Higher age prediction accuracies have previously been reported in several UK Biobank studies, with a particularly strong example by Peng et al. achieving a mean absolute error (MAE) of 2.14 years (<https://doi.org/10.1016/j.media.2020.101871>). How might such improvements in prediction accuracy affect the ability to detect associations between brain-age gap (BAG) and exposome variables? Is it conceivable that, as models continue to improve, their precision could reduce true inter-individual variability in BAG to the extent that biologically meaningful differences become diminished?

[5] Residual confounding by age and sex: The brain-age gap is known to be confounded by age due to regression dilution. The authors regressed out age effects from predicted age using the slope and intercept from the training sample. However, depending on how well the model generalizes to the test sample—even though all data are from the UK Biobank with harmonized scanner protocols—some residual confounding may remain. It would be helpful if the authors explicitly demonstrated that age (and sex) do not confound the BAG–exposome associations, for example by including a plot similar to Supplementary Figure 6.

[6] Sample independence and relatedness: The UK Biobank includes many participants with first-, second-, and third-degree familial relationships. Were related individuals excluded across the training, validation, and test sets to prevent potential data leakage? If not, how did the authors ensure that familial relatedness did not introduce bias into the results? If related individuals were not accounted for, this could represent a methodological limitation worth acknowledging.

[7] Algorithm choice rationale: While ridge regression, SVR, and random forest are valid choices, many other machine learning approaches are available for tabular data. What was the rationale for selecting these three algorithms? Have the authors considered ensemble methods?

[8] P-values across datasets: In the main subset, the exposome model yields $r = 0.23$ with $p = 0.002$. In the larger replication and variable-restricted samples, the correlation remains similar ($r = 0.23$ and 0.24 , respectively), yet the reported p-values are higher. Given the increased sample size, this is unexpected—typically, larger samples lead to smaller p-values for equivalent effect sizes. The authors mention that p-values were FDR-corrected, but it remains unclear which specific tests were included in the correction procedure. I suggest reporting both the raw and FDR-corrected p-values side by side in Supplementary Table 3, possibly along with a brief table note explaining which tests were included in the FDR correction.

[9] Language clarity: The phrase “worst grey matter health” (and similar expressions such as “worst predicted grey matter health”) appears multiple times throughout the manuscript, and I found it a bit unusual. The authors might consider rephrasing these instances using more standard alternatives such as “poorer brain health”.

[10] Sex/gender: Did the authors use self-reported sex or genetically inferred sex in their analyses? Unless a specific gender variable was used, referring to “sex” would be more appropriate in my view. Were discrepancies between self-reported and genetic sex taken into account in the analysis (e.g., by excluding individuals with reported vs. genetic sex-mismatch)?

[11] Interpretation of “almost null gap”: The manuscript states that “most participants showed an almost null gap...,” which may suggest that individual brain-age gaps are negligible. However, the reported mean absolute error is approximately 3.9 years. Do the authors instead refer to the group-averaged mean of the BAG distribution, which may be close to zero by design? I suggest clarifying whether this statement refers to the distribution mean or to individual-level deviations, and consider revising the wording to avoid potential confusion.

[12] GitHub repository: I very much appreciate the inclusion of the GitHub repository, which appears well-organized, clear, and greatly enhances the transparency of the analysis workflow. The authors recommend using virtual environments such as conda, which is good practice. To further improve reproducibility, could the authors provide the corresponding environment.yml file(s) (or similar) with package versions? Additionally, it would be helpful to specify the system used to run the analyses.

[13] The exposome concept: The concept of the exposome was relatively new to me, as I am more familiar with genome- and phenome-focused approaches. As I understand it, the exposome is generally defined as the totality of non-genetic internal and external exposures across the lifespan. However, many of the variables assessed in this study—such as dietary habits, smoking behaviour, or socioeconomic status—are known to have some degree of genetic influence. This suggests that the set of predictors used here may not reflect purely non-genetic exposures in the strictest sense. In my view, this overlap between genetic predisposition and environmental exposure highlights a broader conceptual limitation of the exposome framework. It might be helpful to briefly acknowledge this nuance in the discussion—perhaps in the context of causality, where the exposome framework alone does not resolve the challenge of causal inference, particularly in observational cohorts like the UK Biobank.

(Remarks on code availability)

I briefly reviewed the GitHub repository, and it appears well-organized, well-documented, and comprehensive. To further improve reproducibility, the addition of a conda environment file and basic system information would be beneficial.

Reviewer #2

(Remarks to the Author)

I thank the Editor and Authors for the chance to review this interesting paper! Please find my structured comments below.

Key results

This study presents an interesting approach to answering the question of “What predicts individual brain health?” by using rich data from the UK Biobank. Grey matter-based brain age models are trained and then used to predict brain age in a sub-sample of the UK Biobank. The difference between the individuals’ brain age and chronological age (brain age gap) is used as a proxy for brain health. In a further step, the brain age gap is predicted from a set of variables labelled here as the exposome. The authors present that exposome variable predict between 3%-5% of the variance in the brain age gaps. Using SHAP values, the authors present a ranking of the exposome features in explaining the brain age gap.

Validity

While the manuscript draws on rich and high-quality data, there are multiple uncertainties around BAG still limiting its interpretability. The extent to which BAG reflects brain health remains unclear. Additionally, the results indicate relatively weak associations between the exposome and BAG. These effect sizes were not discussed, which would however be important for context when presenting SHAP values and making conclusions about exposome data being able to predict brain health.

Significance

The integration of exposome data with neuroimaging-based brain age prediction is timely and potentially impactful, particularly in the context of identifying modifiable lifestyle or environmental factors associated with brain health. The use of large-scale population data (UK Biobank) and open methods further enhance the manuscript’s relevance. However, the modest variance explained and the unresolved issues around the interpretation of BAG somewhat limit the strength of the conclusions. While the work is incremental in its current form, stronger positioning within the existing literature and a more critical discussion of the results would increase its contribution to the field. Nevertheless, the findings support a forming body of literature highlighting that the state of the brain (or potentially brain health) is multivariate, being an useful contribution to the field.

Data and methodology

The dataset (UK Biobank) is appropriate and well-regarded. However, important methodological details are missing or underspecified. These include:

- Rationale for the choice of machine learning algorithms.
- Whether training and test subsets were strictly independent.
- The method and covariates used for age-bias correction.
- How exposome variables were selected and grouped (e.g., based on UKB codes or theoretical assumptions).
- Quality control steps for both exposome variables and MRI data.
- Treatment of variable redundancy and imbalanced group sizes across exposome domains.

Additionally, it is unclear why brain age models were evaluated only with MAE and Pearson’s r , whereas exposome-to-BAG models used a broader set of metrics. A consistent evaluation framework would aid interpretability.

Again, while brain age provides a simplified metric of the state of the brain, which can be described as a proxy of brain health, its limitations need to be discussed in order to not mislead readers who are not familiar with the concept.

Finally, I appreciate the authors’ open science approach and well-organized GitHub repository!

Analytical approach

The use of ensemble models such as Random Forests and SHAP value interpretation is appropriate. Nonetheless, the manuscript lacks information on the stability and robustness of the feature importances across methods. Are the SHAP results stable when compared to, for example, permutation importance or bootstrapped variable selection? The choice of performance metrics should also be reconsidered. Reporting R^2 in the main text -- not only in the Supplement -- is essential for gauging practical significance. Moreover, the authors should ensure that statistical tests are labeled clearly; for example, the description of a t-test on a Pearson correlation is technically accurate but might be confusing in context.

Suggested improvements

Most importantly, the conceptual framing of brain age gap (BAG) could benefit from a more cautious and precise treatment. Currently, several statements risk implying that BAG reflects a direct measure of biological ageing, rather than deviation from a normative reference model. It would be helpful to more clearly separate BAG from ageing processes per se, especially in light of recent evidence questioning this link. In line with this, we suggest revising Figure 1 to avoid labeling the x-axis as “true BAG,” which may unintentionally suggest the existence of an objective ground truth. The methodological transparency would also benefit from further detail. The rationale for selecting specific machine learning algorithms should be articulated, along with an assessment of model stability and generalizability. Similarly, please specify how the age-bias correction was implemented—was it linear or non-linear, which covariates were included, and was the slope from the training set used? These details are important for reproducibility and interpretability.

I also recommend expanding the description of the exposome variables, particularly regarding how they were selected, grouped, and whether any steps were taken to mitigate potential imbalance across groups (e.g., lifestyle vs. environmental exposures). Relatedly, it would be helpful to clarify whether weighting or redundancy reduction was applied to address overrepresentation or multicollinearity.

While performance metrics like Pearson’s r are useful, including R^2 values in the main text, rather than relegating them to the supplement, would give readers a more complete sense of the model’s explanatory power. This is particularly relevant given that only a modest proportion of the variance in BAG is explained. These small effects should be clearly stated and considered when discussing the results.

Moreover, the limitations of BAG as a derived and relatively noisy variable should be more directly acknowledged and discussed in a dedicated limitations section.

Finally, some sections of the Discussion could be condensed to reduce repetition and instead used to place the findings in context of comparable studies. Phrasing such as “optimal lifestyle” and broad claims like “all factors...constitute the exposome” could also be softened or made more precise to avoid overinterpretation. Addressing these points would, in my view, substantially enhance the clarity and impact of the manuscript.

Clarity and context

The manuscript is generally well-structured, but several areas require clarification. Some sentences need rephrasing for precision and scientific tone (e.g., “leads to a worst predicted grey matter health”). The Discussion section currently repeats earlier results rather than synthesizing them with broader literature. A more critical and integrative discussion would improve clarity and impact.

References

Multiple references could be added to foster discussion in the Discussion section. This includes both brain age-specific and brain age-unspecific research.

(Remarks on code availability)

The GitHub repository is well organized and the most important bits of the code are available. I did not run the code, but after reading it, I do not see a reason why it would not run (at least after some refinements). Only critique: The code could be commented better, but this is more or less always the case for all code one is not writing oneself.

Reviewer #3

(Remarks to the Author)

In this paper, the authors investigated the association between exotypes and brain age gap. The use of the large UK biobank dataset is a key strength of the study. Further clarification and justification regarding the methodology, reporting and discussion of the results, particularly in the context of existing literature, is required.

- The authors suggest that to avoid potential biases associated with substantial missing values across participants, they focused on variables that were available in at least 2000 participants including both males and females - however, this would introduce bias, as has been well documented via complete cases analysis. Why was a robust method for addressing missing data not conducted, such as multiple imputation?
- It is not clear to me how the statement “BAG does not rely on chronological age” is true, given the brain age gap subtracts the chronological age from the predicted age.
- I do not see how an individualised approach has been adopted here. Fundamentally, these models show relationships between a set of variables and brain age gap, but do not provide an individualised approach to predicting brain age gap.
- At what timepoint were the exposures measured, in addition to the imaging data? Was the time between measures considered in the model?
- The entire discussion is missing a meaningful discussion of effect sizes i.e., how many additional years in brain ageing were observed as a result of each additional year for the exposure? This is needed to provide a meaningful interpretation of the results.
- It is important that all statements are supported by citations/evidence
- Please include a descriptive table for variables used in study

(Remarks on code availability)

Code is well documented

Version 1:

Reviewer comments:

Reviewer #1

(Remarks to the Author)

The authors responded to my revision requests with notable rigor and thoroughness, which substantially improved the manuscript. All my previous concerns have been fully addressed. I have only two remaining comments / tips for finalization.

Relatedness field: Unfortunately, the tabular data fields referring to relatedness, as mentioned by the authors, are not very informative. There is a file containing pre-calculated kinship coefficients that could previously be downloaded using the UK Biobank tool `gfetech` with the `-rel` flag. Within the Research Analysis Platform, this relatedness file including kinship coefficients can now be found under `Bulk/Genotype Results/Genotype calls/ukb_rel.dat`. Possibly something that could be considered in a future study. However, access to this file may require a Tier 3 plan.

P-values and permutations: Thank you for the clarification. With 500 permutations, the lowest achievable p-value is < 0.002 (i.e., when none of the permutations yields a similarly extreme test statistic). It may be worth mentioning this explicitly, as it

explains why many p-values appear as 0.002. I also noticed that for SVR (linear) and r-squared, the corrected p-value is lower than the uncorrected p-value; please re-check this. Additionally, consider reporting all p-values in this table with three decimal places for consistency.

I congratulate the authors on this strong revision.

Best wishes,
Philippe Jawinski

(Remarks on code availability)

The GitHub repository is well-organized, well-documented, and comprehensive. As requested, the authors have added a conda environment file during the revision, which further enhances reproducibility.

Reviewer #2

(Remarks to the Author)

The authors have addressed my comments satisfactorily. Congratulations on a well-executed study!

(Remarks on code availability)

The authors have addressed my comments satisfactorily. Congratulations on a well-executed study!

Reviewer #3

(Remarks to the Author)

The authors have made considerable progress in addressing the initial comments and should be commended for this. However, further clarification and justification is required for transparency.

- The authors' rationale for not conducting multiple imputation is understandable, but this justification must be included directly in the manuscript. This is important because the decision not to impute has implications for potential bias. A concise explanation of the missingness structure, the limitations of MI under these conditions, and the trade-offs involved should be made clear to readers.

- The response regarding the "individualised approach" is appreciated, but the manuscript should more clearly distinguish between what the current model can achieve and what may be possible in future applications throughout. At present, the predictive accuracy is not sufficient to support individual-level inference, and any statements implying personalised prediction should be reframed to refer explicitly to future potential rather than current capability.

- The authors state that "small variations in the time between exposome measurement across subjects is unlikely to influence the pattern of results," but this assumption needs to be transparently stated in the manuscript with the justification provided, as there is currently no evidence to support this.

- The discussion of effect sizes remains limited. While classical regression-style effect estimates cannot be derived from the modelling approach used, readers still require a meaningful way to interpret the magnitude and direction of key associations. The authors are encouraged to add brief clarifying text in the discussion noting these limitations, and include SHAP dependence plots or similar visualisations to illustrate how the most influential exposures relate to BAG across their value ranges.

(Remarks on code availability)

Have not re-reviewed code.

Dear Editors and Reviewers,

Thank you for your letter and for the reviewers' comments concerning our manuscript entitled "What predicts individual brain health? a machine learning study spanning the exposome". Your comments were highly insightful and enabled us to greatly improve the quality of our manuscript. We have studied comments and made corrections carefully which we hope to meet with approval. In the following pages are our point-by-point responses to each of the comments of the reviewers. Revisions in the text are shown using blue colour.

Responses to the comments of Reviewers:

Reviewer #1 (Remarks to the Author):

In their study, Mahdipour et al. seek to shed light on how a broad spectrum of biomedical, lifestyle and psychosocial exposures jointly relate to individual differences in "brain age gap" — the discrepancy between an individual's age predicted from structural MRI data and their chronological age. Here, brain-age gap is used as a proxy for overall brain health. The authors implement a two-step pipeline using data from the UK Biobank: (i) three machine learning algorithms (ridge regression, support vector regression, and random forest) are trained on grey-matter parcels from 5,025 healthy adults to predict brain age in 34,365 independent participants; (ii) a subset of these participants, scored on up to 261 exposome variables spanning cardiovascular-metabolic, behavioural, socio-economic, and early-life domains, are used to predict the brain-age gap via a random forest model, with SVR and ridge regression used as sensitivity analyses. Model performance is assessed using nested cross-validation, and feature importances are quantified using SHAP values. The exposome explains approximately 5% of the variance in brain-age gap ($r \approx 0.23$, $p = 0.002$). Key contributors include alcohol intake, cardiometabolic burden (e.g., blood pressure history, age at diabetes onset), smoking behaviour, musculoskeletal factors, and dietary patterns (e.g., coffee consumption), while mental health, socio-affective, and environmental variables contribute comparatively little. The authors conclude that modifiable lifestyle and metabolic exposures modestly—but reliably—affect overall brain health, emphasizing the importance of early prevention in cardiovascular and metabolic domains to promote healthy brain ageing.

The manuscript is well-written, methodologically sound, and the analyses are logically organised. I particularly commend the authors for releasing their full codebase in a public GitHub repository, which greatly enhances transparency and reproducibility. The topic is timely and relevant to researchers in neuroscience, ageing, and epidemiology. A notable strength is the study's explicit attempt to disentangle the individual contributions of a broad range of exposures to brain health, thereby illuminating modifiable risk factors. Nevertheless, the work's capacity for causal inference is limited, and several of the associations—though now examined in a multivariate framework—have been reported previously in similar UK Biobank analyses. That said, the work still adds valuable evidence and provides timely insights into modifiable influences on brain health. In the following, I outline a number of concerns that, if addressed, would further strengthen the manuscript.

We thank the reviewer for the positive evaluation of our manuscript, as well as his/her comments which have contributed to improve its quality.

[1] Novelty and prior work: A broad spectrum of phenotypic associations between lifestyle, health factors, and brain ageing have already been explored in UK Biobank data (e.g., Smith et al.: <https://doi.org/10.1016/j.neuroimage.2019.06.017>; Jawinski et al.: <https://doi.org/10.1101/2023.12.26.23300533>). How do the current findings extend or differ from

these prior studies? Please clarify how the present multivariate approach advances beyond earlier univariate correlation analyses.

We thank the reviewers to give us the opportunity to discuss the novelty of our work, in particular when compared to previous phenome-wide association studies of BAG, such as Smith et al. (2019)¹ and Jawinski et al. (2023)². The main difference lies in the approach: while phenome-wide studies typically test association with BAG for each phenotypical variable separately, our exposome holistic approach considered all variables conjointly with a predictive modelling accounting for redundancy and interaction.

A consequence of the redundancy or collinearity among exposome variable in a mass bivariate testing as done in Smith et al. (2019)¹ and Jawinski et al. (2023)² is that all variables that are colinear will appear as significantly associated with BAG. Consequently, there is no indication whether one might be more predictive for BAG than the other related variable and this may lead the authors to make subjective decision on how to filter the set of related variables for reporting and interpretation. For example, in Smith et al., the authors reported “we have manually removed largely-redundant variables for purposes of readability”’ and accordingly they listed only the strongest linear association (correlation) for a set of similar variables. Despite this statement, the authors still report many related variables in their main results, such as “Alcohol intake frequency, Average weekly beer plus cider intake and average weekly red wine intake”. In contrast, our approach shows that “Alcohol intake” on its own contribute to predict individual brain health and that the type of alcohol does not really play a role in the prediction.

Another consequence of bivariate mass testing is that interaction effects are completely ignored potentially missing relevant modulating factor. We argue that this is the reason why some factors appear specifically relevant when predicting individual brain health with a multivariate predictive modelling approach in our study. In particular, coffee intake and dried fruit intake did not appear as key factors in previous phenome-wide association studies. Furthermore, hip circumference was also rarely reported in association to brain health and was not linearly associated to BAG in our study, although it appears to play an important role in the prediction of individual brain health. Therefore, we now further discuss the role of these variables in interaction with other factors. In particular, we discuss the protective role of gluteofemoral fat in cardiovascular and metabolic conditions, as well as the protective role of dried fruits intake on cardiovascular conditions based on evidence from previous studies. In addition, we also discuss potential detrimental interaction between coffee intake and cardiovascular factors (such as blood pressure).

Finally, but very importantly, we would also like to point out that our study for the first time points to the relevance of information pertaining to the window and duration of detrimental exposures, such as smoking, high blood pressure and diabetes. To the best of our knowledge, this aspect has rarely been discussed before, while according to our model the time period of exposure play an important role in determining individual brain health.

Following the reviewers’ comments, the novelty of our work has been further discussed in the revised manuscript:

In the introduction (page 3, line 7-15):

“Using a BAG-based grey matter health estimator, some studies have emphasized that grey matter health is tightly linked to several aspects of body health in aging populations³⁻⁵, others have highlighted relationships with early life factors⁶ and socio-affective/mental health-related factors (e.g., long-term depressive symptoms⁷). Given the wide variety of factors that can influence brain health, some studies have applied a phenome-wide association analysis of BAG^{1,8}.

With such approach, for each and every available non-genetic variable, its association with BAG is tested separately, typically for a linear association. Such approaches hence do not account for redundancy and interaction among different factors altogether forming the expotype of an individual. These previous studies may thus have missed important factors appearing only in interaction with others in shaping brain health in aging.”

In the discussion (page 8, line 6-18):

“Altogether these results generally point to the fact that the major threats for brain health in our aging populations are metabolic and cardiovascular factors.

However, in contrast to previous phenome-wide association studies of BAG^{1,8}, our multivariate predictive modeling approach allows us to identify that the top contributing variables for the key other risk factors that are smoking, high blood pressure and diabetes pertain to the life period at which the exposure has taken place and the duration of the exposure. For smoking and higher blood pressure, we can even note that both, variables pertaining to life period, and variables pertaining to duration conjointly contribute to the prediction. Hence our multivariate predictive modeling approach allows us to demonstrate that life period and chronicity of the exposure are complementary (rather than redundant) information for determining individual brain health. Overall, the sooner these detrimental exposures are discarded (by stopping smoking and by diagnosing and treating high blood pressure), the lower the negative impact on brain health. Thus our study crucially highlights the need of early prevention campaigns⁹ targeting diabetes, smoking and blood pressure monitoring for public health strategies.”

(page 9, line 3-9):

“However, previous phenome-wide association studies did not highlight coffee consumption as strongly associated with BAG when taken in isolation^{1,8}. This suggests that the detrimental effect of high coffee consumption on brain health can potentially be explained by interaction with other risk factors discussed above, in particular with cardiovascular factors. It has for example been showed that in UK Biobank participants with genetic predisposition to elevated intraocular pressure, greater caffeine consumption was associated with higher intraocular pressure and higher glaucoma prevalence¹⁰.”

(page 9, line 19-24):

“Furthermore, dried fruits intake have been demonstrated to play a protective role against cardiovascular disorders in UK Biobank data¹⁴. Although further studies are needed in the future to better understand how diet and other factors interact with metabolic and cardiovascular factors to influence individual brain health, our current study suggests that high (whole-grain)

cereal and dried fruits intake are important factors contributing to promote grey matter health in aging.”

(page 9, line 25- page 10, line 8):

“In our machine learning models, hip circumference also appears to contribute to the prediction of individual grey matter health. More concretely, lower (than the mean of 102 cm, see Supplementary table 10) hip circumference is associated with poorer individual brain health in the prediction model. However, additional analyses show that, when taken in isolation, hip circumference is not associated with grey matter health in the main subset (null correlation with BAG, see Supplementary Figure 6). This suggests that hip circumference plays a role as a moderator variable in the individual prediction of grey matter health. In other words, lower hip circumference on its own does not result in poorer brain health, but in interaction with other factors, it can lead to poorer predicted brain health. It can for example be assumed that a combination of low hip circumference and low bone density reflects physical frailty, a body condition associated with neurocognitive impairment¹¹. In this vein, when hip circumference is taken in conjunction with other variables, it can indicate specifically local adiposity, which can have a protective effect against some metabolic and cardiovascular conditions. In particular, hip circumference is an indicator of gluteofemoral fat which, in contrast to visceral fat, is associated with reduced risks of insulin-resistance, type-II diabetes and cardiovascular problems, as well as, to metabolic health^{12,13}. Since we observe here that hips circumference, a proxy for gluteofemoral fat, does contribute to predict grey matter health while it is not directly linearly associated to grey matter health in the sample, we can assume that gluteofemoral fat may modulate the influence of cardiovascular and metabolic risks factors on brain degeneration through complex pathways. This is an interesting question for future studies which should determine exactly how hips circumference/gluteofemoral fat modulates the relationship between cardiovascular-metabolic health and brain structural health.”

(page 11, line 20-25):

“Despite these limitations, the current study revealed a set of factors primarily contributing to predict structural brain health in an aging population, highlighting cardiovascular health, bone health and diabetes as key factors, confirming previous studies based on univariate, linear and not cross-validated modeling^{1,8,15}. More importantly, our study pointed to additional factors, which may not show strong linear association to brain structural health when taking in isolation and which therefore are assumed to interact with other factors, such as nutrition diet and coffee intake, as well as hip circumference.”

[2-A] Variable selection and interpretation: It remains somewhat unclear how the 261 exposome variables were selected. Were these variables chosen purely based on availability and relevance, or were

they also selected to represent independent clusters of health-related domains? How distinct are the included variables from one another in terms of conceptual overlap and statistical collinearity? Providing a hierarchical clustering analysis or correlation heatmap could help demonstrate the structure and independence of the variable set.

In short, we took a two-step process: we first adopted a top-down approach to select relevant exposome domains and then constrain the variable set based on availability. Accordingly, the variables were mainly selected to represent independent clusters of health-related domains, based on the literature in this field.

To clarify this point, below we provide a detailed explanation of how the exposome variables were selected:

- **Literature:**

We reviewed more than 50 scientific articles, mainly in two categories:

- *Studies that investigated associations between Brain Age Gap (BAG) and exposome variables.*
- *Studies that investigated associations between brain structure/brain health and exposome variables.*

Some studies were theory-driven and focused on a single variable or category. For example, Cherbuin et al., 2021¹⁶ investigated the relationship between group of blood pressure (BP) variables and brain health, as measured by BrainAGE, over a 12-year period in middle-aged and older adults; or Gray et al., 2020¹⁷ assessed the relationship of cigarette smoking with gray matter (GM) and white matter (WM) in the UK Biobank, controlling for numerous confounding demographic and health variables.

Other studies used an agnostic approach (phenome-wide association) or considered a wide range of variables. For instance Miller et al., 2016¹⁸ investigated the relationship between 1100 non-imaging derived phenotypes (coming from 378 data fields) and 2501 imaging derived phenotypes (IDPs). In the brain age context, Smith et al., 2020¹⁹ and Dinsdale et al., 2021²⁰ used 8787 non-imaging derived phenotypes (coming from more almost 500 data fields) to investigate the relationship of BAG and other phenotypical variables. Similarly, Tian et al., 2023³, by using UK Biobank imaging data, designed 3 brain and 7 body (organs) age models. They have done extensive work on investigating the relationship of the gap between brain and body model. They have also investigated the associations between organ-specific age gaps, including brain age gap, and several environmental/lifestyle measures. They have used 148 (non-imaging) Field IDs as exposome variables. These exposome variables include smoking, alcohol consumption, socioeconomic inequality, air pollution, greenspace, natural environment coverage, exercise, education, sleep hygiene, and maternal nutrition.

- **Exposome variables:**

Next, we listed all variables reported in previous literature and accessible in UK Biobank for our study. For previous studies conducted in UK Biobank, we could straightforwardly identify the variables based on field IDs. For studies based on other cohorts, we mapped the exposome variables used there to equivalent variables in UK Biobank (where possible). After initial listing, we have done a preliminary exclusion to exclude variables not pertaining to the exposome or that were less relevant duplicated of another field. Concretely, we have excluded the following categories and Instance IDs:

- *Mother category of 100026 (Cognitive function) and all her children's categories*
- *Mother Category of 116 (Cognitive function online) and all her children's categories*
- *Instance ID number 1 (Name: Diet Questionnaire Cycles): We excluded Instance ID 1, which corresponds to 'diet questionnaire cycles', due to its focus on daily food consumption rather than overall dietary patterns. Instead, we utilized variables from*

the broader Category 100052 (Diet), which provides information on the frequency of food consumption, aligning better with our study's focus on long-term dietary habits. This approach allowed us to focus on variables that reflect long-term dietary patterns, which are more relevant to our study of Brain Age Gap.

- Instance ID 12 (Name: Coronavirus serology study wave)
- Instance ID 27 (Name: Coronavirus serology study symptoms questionnaire)
- Instance ID 93 (Name: Accelerometer wear)
- Instance ID 154 (Name: Covid-19 antibody study phase)
- Instance ID 178 (Name: Online cognitive assessment)
- Instance ID 604 (Name: Vaccination event)
- Instance ID 9000001 (Name: Death registry reports)
- Instance ID 9000002 (Name: Cancer registry reports)

In total, we compiled 659 Field IDs across 67 categories. From these, we selected 397 Field IDs, prioritizing those most frequently used in the literature. These included 291 common fields (39 categories) with the literature covering major domains such as body morphology and composition, diet, lifestyle, smoking, alcohol consumption, and cardiovascular health. The remaining 106 fields were mainly related to body injuries, illnesses, and socio-affective aspects.

- **Parsing data from UK Biobank:**

As described in the Methods section of the manuscript, we used the `ukbb_parser` Python package Brandes et al., 2020²¹ to extract data for the 397 Field IDs. This resulted in a table with 2,599 columns, because each Field ID may include multiple time instances (e.g., baseline and follow-up), and categorical fields with multiple responses are expanded into even more columns. Thus, the number of columns substantially exceeded the initial number of Field IDs.

Using the `Missingno` package, we examined missing data patterns (Figure 1), which highlighted substantial missingness as a major challenge in selecting Field IDs as exposome features.

Figure 1. Missingness pattern in the literature-based a-priori selected data fields (`Missingno` package). The grey colours are the filled values in women participants. Blue colours are the filled values in men participants. White colour reflects missing values, and red colour indicates non-informative answers (i.e., I don't know, I prefer not to answer etc.).

- **Data curation:**

In the next step, we curate the data to ensure validity and meaningfulness:

1. **Handling missing values:** We retained columns with at least 2,000 filled values.
2. **Variable type (categorical vs. continuous) assignment:** We identified which features were categorical (discrete groups) and which were continuous (numeric measurements), to apply the appropriate preprocessing and modelling strategies.
3. **Uniqueness constraint:** We ensured that each field ID (and participant) was represented uniquely in the dataset and removed potential duplicate records.
4. **Aberrant value exclusion:** We verified that all variable values fell within the valid range or allowed set of categories (e.g., no out-of-range numerical values or invalid category codes).
5. **Uninformative value exclusion:** We excluded implausible or non-informative responses. For example, some participants selected “do not know” or “prefer not to answer,” which were removed from further analyses.

- **Calculation of Durations:**

In the final step, there were some variables from medical conditions and medical information (body injuries and illnesses) category, that they were related to the age when a disease happened. Subtracting these ages from the age when attended assessment centre (chronological age) provided the duration of those illnesses.

At the end, and considering all mentioned steps, this led us to define three subsets. As mentioned in the manuscript, in the main subset 261 distinct exposome variables were available for 3706 participants. By dropping out two variables (left and right heel bone density), we could create a bigger subset of 4292 participants that served as a replication subset. Finally, we also identified a bigger subset of 7736 participants with data for 201 distinct exposome variables.

Additional information regarding these steps has now been provided in the revised manuscript:

(page 15, line 6-29):

“Exposome variables

To represent the exposome, we used a wide range of non-imaging variables encompassing biomedical, lifestyle, socio-economical and early life factors as illustrated in Figure 1 and the full list of exposome variables are available in Supplementary Table 2. The selection of variables was guided by previous publications reporting either associations (with correlation or regression usually) with brain health as measured by the brain age gap^{5,6,15,22-27} or associations with grey matter structure^{18,28-32} in the UK Biobank. This choice was also constrained by data availability. To avoid potential biases associated with substantial missing values across participants, we focused on variables that were available in at least 2000 participants including both males and females. We also performed some additional data curation processes, which included: uniqueness constraint (we ensured that each field ID and participant was represented uniquely in the dataset and we removed potential duplicate records), aberrant value exclusion (we verified that all variable values fell within the valid range or allowed set of categories e.g., no out-of-range numerical values or invalid category codes), and uninformative value exclusion (we excluded implausible or non-informative responses such as “do not know” or “prefer not to answer”).

Furthermore, we also calculated duration variables for medical conditions by subtracting onset age from the age when attended assessment center (chronological age).

As illustrated in Figure 1, these procedures led us to define a main subset in which 261 distinct exposome variables were available for 3706 participants. By dropping out two variables (left and right heel bone density), we could create a bigger subset of 4292 participants that served as a replication subset. Finally, we also identified a bigger subset of 7736 participants with data for 201 distinct exposome variables. In this “variables-restricted subset”, mainly variables related to socio-affective and mental health domains were missing compared to the main and replication subsets. Standard preprocessing was performed on exposome variables including variables standardization (within the cross-validation scheme to avoid any data leakage).”

[2-B] Regarding variables collinearity and potential redundancy (“How distinct are the included variables from one another in terms of conceptual overlap and statistical collinearity? Providing a hierarchical clustering analysis or correlation heatmap could help demonstrate the structure and independence of the variable set”):

To address the reviewer's query regarding the distinctiveness and potential collinearity of the 261 exposome variables, we have generated a correlation heatmap and performed a hierarchical clustering analysis.

The hierarchical clustering analysis (see Figure 2 below, a high-resolution PDF file has also been uploaded with the revisions file) provides a visual representation of the relationships between the variables. As can be seen in the dendrogram, the variables cluster into distinct groups generally aligning with broad categories such as diet, body composition, cardiovascular health and socio-affective domain. This suggests that while some variables within these categories may be correlated, the overall variable set captures a diverse range of exposome domains. The branches show how features are related based on distance.

Figure 2 Hierarchical clustering: Using pre-processed features (exposome variables) from the main subset, a hierarchical clustering was performed. The variables' names have been colour-coded by domain.

The correlation heatmap (see Figure 3 below and Supplemental Figure 8, a high-resolution PDF file has also been uploaded with the revisions file) provides a more detailed view of the pairwise correlations between variables. Only a few clusters of more correlated variables are apparent (red blocks), and these pertain to body composition and body morphology variables, which could be expected to be importantly correlated. Apart from these clusters, overall, the heatmap shows a substantial degree of heterogeneity, indicating that many variables exhibit relatively low correlations with one another. This suggests that the variable set, while capturing some expected relationships, also includes a good degree non-redundant, distinct information piece. We did not perform explicit redundancy reduction, as the Random Forest algorithm is inherently robust to multicollinearity and capable of selecting the most informative variables. We believe the heatmap sufficiently illustrates any inherent redundancy within the variable set, without the need for further pre-processing steps.

Figure 3 (Supplemental Figure 8). correlation heatmap of the preprocessed features. Clusters of more correlated variables pertain to body composition and body morphology variables

Following the reviewer's comment, this information has been included in the revised manuscript (page 15, line 29- page 16, line 2):

“The redundancy between the 261 exposome variables were relatively low as illustrated by the correlation heatmap (Supplementary Figure 8). Although some clusters of correlated variables can be observed, these pertain to instrumentally related body morphology and composition variables and overall, a substantial degree of heterogeneity can be observed. The distribution of the variables across different exposome domains is illustrated for reader's information in Supplementary Figure 9. Of note, from this set of variables, we did not perform any redundancy reduction since we subsequently used machine learning algorithms which are designed to be able to weight or select relevant features (see next section).”

[3] Training sample characteristics: In lines 120-125, the age range is reported for the test sample, but not for the training sample. For clarity and consistency, it would be helpful to also provide the age range of the training sample, ideally in the same section.

We appreciate the reviewer's attention. To improve clarity and consistency, we added the age range of the training sample to the Results section, in addition to its current location in the Participants section (Method section). The revised sentence reads (page 4, line 8-11):

“To build an estimator that would later be sensitive to deviance from a healthy reference norm, the models were trained and explicitly tested in a subset of 5025 healthy participants from the UK Biobank (age range 46-82 years, mean 62.12±7.16 years, 2579 females, see Methods).”

[4] Prediction accuracy and implications for BAG: Higher age prediction accuracies have previously been reported in several UK Biobank studies, with a particularly strong example by Peng et al. achieving a mean absolute error (MAE) of 2.14 years (<https://doi.org/10.1016/j.media.2020.101871>). How might such improvements in prediction accuracy affect the ability to detect associations between brain-age gap (BAG) and exposome variables? Is it conceivable that, as models continue to improve, their precision could reduce true inter-individual variability in BAG to the extent that biologically meaningful differences become diminished?

We appreciate the reviewer's point regarding the potential impact of increased brain age prediction accuracy on the brain-age gap (BAG) and its relationship with exposome variables. We acknowledge that Peng et al³³ achieved a higher brain age prediction accuracy (MAE = 2.14 years) in the UK Biobank compared to our model (MAE = 3.93 years). This difference likely arises from methodological choices, including our selection criteria for a healthy training set (resulting in a smaller training sample), our focus on gray matter volume features to have a strict indicator of grey matter health, and the specific regression algorithms employed.

In this study, we aimed for relevant indicator of grey matter health in the population instead of a highly accurate brain age prediction model. The reason for this choice is, as stated by the reviewer, that model accuracy does not imply, and can even result in lower, sensitivity for interindividual variability in brain health. This counterintuitive fact about brain age prediction accuracy has been recently demonstrated in UK Biobank³⁴.

In line with this fact, for the current study, as brain age prediction models improve, the distribution of BAG values will likely become narrower, reflecting a reduction in unexplained variance. This refinement of BAG has implications for detecting associations with exposome variables. Specifically, with a narrower BAG distribution, the ability to detect relationships may be limited to exposome features that exhibit stronger correlations or larger effect sizes on brain aging. In other words, the 'signal' from weaker associations may be overshadowed by the increased precision of the BAG measurement. Furthermore, while Peng et al.'s³³ model demonstrates impressive overall accuracy, our gray matter-focused model may be more sensitive to the specific effects of certain exposome variables on grey matter alterations.

In conclusion, there is a trade-off between model accuracy and BAG variance. The optimal balance between these factors depends on the specific research question. Furthermore, the extent to which the information conveyed by BAG depends on how the brain age prediction model has been developed remains a field of study. Accordingly, and in line with other reviewers' comment, we have now acknowledged current limitations in the validity and interpretation of BAG as an index of brain aging and brain health in the revised manuscript:

In the discussion (page 11, line 4-12):

“Second, it should also be acknowledged that although BAG is a very convenient indicator of the degree of relative preservation or degeneration of a whole grey matter pattern, it remains a derived indicator instead of a direct measurement, which implies some noise. Furthermore, its conceptual validity and interpretability remain importantly discussed³⁵. First, a BAG framework assumes a unique aging trajectory and deviance (older brain age) from this trajectory is considered as advanced or pathological aging³⁵. Yet, this assumption can be questioned. Second and relatedly, some studies suggest that deviance (older brain age) may reflect brain phenotype present since early life instead of an aging phenotype³⁶. Overall these considerations question the extent to which BAG can be used as an indicator of brain health in aging.”

[5] Residual confounding by age and sex: The brain-age gap is known to be confounded by age due to regression dilution. The authors regressed out age effects from predicted age using the slope and intercept from the training sample. However, depending on how well the model generalizes to the test sample—even though all data are from the UK Biobank with harmonized scanner protocols—some residual confounding may remain. It would be helpful if the authors explicitly demonstrated that age (and sex) do not confound the BAG–exposome associations, for example by including a plot similar to Supplementary Figure 6.

We thank the reviewer for raising this important point. In this project, we took great care to ensure that results are not confounded by age. We addressed this issue first by applying a bias correction to the predicted age using the slope and intercept derived from the training set. As suggested by the reviewer, we examined the correlation between BAG and age before and after this correction (see Table 1/Supplementary Table 11). While a very small correlation remains ($r = -0.123$, -0.135 , and -0.130 for the Variables-restricted subset, Replication subset, and Main subset, respectively), it represents a substantial reduction compared to the correlation before correction ($r = -0.668$, -0.674 , and -0.672 , respectively). To further mitigate any potential residual confounding effects in the BAG-exposome associations, as well as some potential confounding effect of sex, we included age, age² and sex as confounds within the cross-validation scheme in the exposome-based prediction of BAG.

Table 1 (Supplementary Table 11) Correlation Between Chronological Age and Brain Age Gap (BAG) Before and After Bias Correction

	MAIN	REPLICATION	VARIABLES-RESTRICTED
CORRELATION (R) BEFORE CORRECTION	-0.672	-0.674	-0.668
CORRELATION (R) AFTER CORRECTION	-0.130	-0.135	-0.123
SCATTER PLOT (CHRONOLOGICAL AGE VS. BAG)			
Note: Pearson's correlation coefficients (r) between chronological age and brain age gap (BAG) in different subsets of the UK Biobank cohort, before and after applying bias correction to the predicted age. Scatter plots illustrating the relationship between chronological age and BAG are also shown. The bias correction was performed using the slope and intercept derived from the training set.

Age Bias correction in the methods (page 14, line 12-22):

“Importantly, several studies have brought attention to an age bias/age dependency in brain age prediction requiring a so-called age-bias correction³⁷. This was done here by regressing out the effects of age on the predicted age³⁸. Specifically, we calculated a regression line with chronological age as the predictor and predicted brain age as the outcome, using only the training dataset. The resulting slope and intercept were then used to adjust the predicted brain age values in the testing dataset. This adjustment involved subtracting the intercept from each predicted brain age and then dividing by the slope. In other words, the slope and intercept of the regression line between chronological age and the predicted age were estimated in the training set and used to adjust the predicted age in the test and the population samples. As expected, and illustrated in supplementary Table 11, this age-bias correction procedure substantially reduced the artefactual correlation between the subsequently computed BAG and age.”

Age and sex as confounders in exposome-based predictive model of BAG (page 16, line 21-26):

“Controlling for covariates was performed on the exposome variables within the cross-validation framework (using estimated from the train set). The set of covariates/ confounds include age, square of age, sex, height, and volumetric scaling from T1 head image to standard space. Thus, all care was taken to neutralize the influence of usual confounds by using a normative indicator

of grey matter health and by additionally controlling for covariates within the prediction analysis.”

[6] Sample independence and relatedness: The UK Biobank includes many participants with first-, second-, and third-degree familial relationships. Were related individuals excluded across the training, validation, and test sets to prevent potential data leakage? If not, how did the authors ensure that familial relatedness did not introduce bias into the results? If related individuals were not accounted for, this could represent a methodological limitation worth acknowledging.

We thank the reviewer for drawing our attention to this point. We acknowledge that our analysis did not explicitly exclude or account for related individuals across the training, validation, and test sets. The presence of related individuals could potentially slightly influence the prediction performance as family members may share similar genetic and environmental characteristics, leading to non-independent observations.

To the best of our knowledge and potentially due to data protection protocols, there is no directly accessible field ID in the UK Biobank that definitively identifies familial relationships for all participants. We have identified genomic-related field IDs that provide some information on genetic relatedness: 22011 (Genetic relatedness pairing), 22012 (Genetic relatedness factor), 22013 (Genetic relatedness IBS0), and 22018 (Genetic relatedness exclusions). However, these fields are only calculated for a subset of 17,296 participants in our dataset, which includes over 35,000 individuals. Therefore, we acknowledge that not accounting for familial relatedness represents a limitation of our study. This limitation has now been stated in the revised manuscript (page 10, line 22-25):

“However, several limitations should be noted. First, at the methodological level, in this study we did not account for family relatedness. The UK Biobank includes several participants with first-, second-, and third-degree familial relationships which can potentially bias the results.”

[7] Algorithm choice rationale: While ridge regression, SVR, and random forest are valid choices, many other machine learning approaches are available for tabular data. What was the rationale for selecting these three algorithms? Have the authors considered ensemble methods?

We thank the reviewer for raising this point. When selecting the algorithms, we took into account brain age prediction literature and complementary among algorithms. Thus we selected ridge regression, support vector regression (SVR), and random forest (RF) due to their established use and effectiveness with tabular data in the field of brain age prediction³⁹⁻⁴¹. Using these well-established algorithms, as their performances are comparable⁴¹, allow us to compare our brain age prediction results with other studies in the field. We also focused on those three algorithms for exposome-based prediction of brain health because they offer complementary strengths: ridge regression provides regularization to prevent overfitting, SVR is very popular and relatively more robust to outliers than ridge regression, and random forest as an ensemble method, has the ability to capture non-linearity and complex interaction among features, furthermore by combining multiple decision trees it usually improves prediction accuracy and robustness. This information has now been added in the revised manuscript:

(page 16, line 10-14):

“We selected these three algorithms because they offer complementary strengths: random forest as an ensemble method, has the ability to capture non-linearity and complex interaction among

features. Furthermore, by combining multiple decision trees it usually improves prediction accuracy and robustness. While ridge regression provides regularization to prevent overfitting, SVR is very popular and relatively more robust to outliers than ridge regression.”

[8] P-values across datasets: In the main subset, the exposome model yields $r = 0.23$ with $p = 0.002$. In the larger replication and variable-restricted samples, the correlation remains similar ($r = 0.23$ and 0.24 , respectively), yet the reported p-values are higher. Given the increased sample size, this is unexpected—typically, larger samples lead to smaller p-values for equivalent effect sizes. The authors mention that p-values were FDR-corrected, but it remains unclear which specific tests were included in the correction procedure. I suggest reporting both the raw and FDR-corrected p-values side by side in Supplementary Table 3, possibly along with a brief table note explaining which tests were included in the FDR correction.

We thank the reviewer for raising this important point regarding the p-values in Supplementary Table 3. We fully agree with the reviewer’s point that p-values tend to decrease with increasing sample size, even for similar effect sizes. However, we would like to clarify that the p-values reported in Supplementary Table 3 were obtained using permutation testing, and we used 500 permutations for all datasets. This means that the number of permutations—and therefore the basis for estimating the p-values—was identical across the three datasets, regardless of their sample sizes.

As requested, we have updated the Supplementary Table 3 (Table 2 here, Supplementary Table 5 in revised version) to report both the raw p-values and the FDR-corrected significance. One clarification we would like to emphasize is that in the previous version of the table, we had reported raw p_values while their significance was determined based on FDR-corrected p_value threshold. In the revised table, we now provide the actual raw p-values corresponding to those tests, along with the FDR-corrected results, for transparency.

Table 2 (Supplementary Table 5). Grey matter health prediction performance across different subsets/algorithms including main subset (n = 3706), replication subset (n = 4202), and variable-restricted subset (n = 7736).

Algorithm	Random Forest			SVR(Linear)	Ridge regression
Subset	Main	Replication	Variables-restricted	Main	Main
Mean Absolute Error (MAE)	4.48	4.52	4.53	4.54	4.53
Uncorrected p-value	0.03	0.02	0.002	0.002	0.02
Corrected p-value	0.03	0.02	0.003	0.003	0.003
Mean Squared Error (MSE)	32.22	32.63	32.66	33.14	33.07
Uncorrected p-value	0.006	0.02	0.002	0.002	0.002
Corrected p-value	0.007	0.02	0.003	0.003	0.003
root Mean Squared Error (rMSE)	5.67	5.71	5.71	5.75	5.75
Uncorrected p-value	0.004	0.04	0.002	0.002	0.002
Corrected p-value	0.005	0.04	0.003	0.003	0.003
r-square	0.05	0.05	0.06	0.02	0.03
Uncorrected p-value	0.002	0.004	0.002	0.02	0.002
Corrected p-value	0.004	0.01	0.002	0.016	0.002
Pearson Correlation (r)	0.23	0.23	0.24	0.16	0.17
Uncorrected p-value	0.002	0.004	0.002	0.002	0.002
Corrected p-value	0.003	0.005	0.003	0.003	0.003

[9] Language clarity: The phrase “worst grey matter health” (and similar expressions such as “worst predicted grey matter health”) appears multiple times throughout the manuscript, and I found it a bit unusual. The authors might consider rephrasing these instances using more standard alternatives such as “poorer brain health”).

We agree with the better phrasing suggested by the reviewer. We have now replaced all instances of phrases like 'worst grey matter health' with more standard alternatives such as 'poorer brain health'.

(page 6, line 1-23):

“Additional insight into the nature of the association between these most contributing variables and grey matter health can be obtained by examining the distribution of SHAP value in Figure 3. This figure illustrates how the value (from low to high) of the exposome variable is associated with the BAG prediction (from negative which corresponds to better grey matter health to positive which corresponds to poorer brain health). Overall, most of the associations follow an expected pattern.

Concretely, for factors that are consensually considered as risk factors for brain diseases, higher exposure is associated with poorer brain health (i.e. higher BAG). These include high blood pressure, smoking and alcohol, as well as diabetes. It should be noted here that for these domains, the variables that show the highest impact on the predicted grey matter health pertain to the duration of the exposure and the life periods in which the exposure happens. Hence, for smoking, duration, age at start and age at stop appear as complementary information instead of redundant variables for the prediction. Longer smoking duration, younger age when started smoking and older age when stopped smoking are all associated with poorer predicted brain health. Similarly for high blood pressure, a longer duration and an older age at diagnosis (likely reflecting an older age for treatment) both lead to poorer predicted brain health. Along the same line, our results show that when diabetes diagnosis happens at an older age, a poorer brain health is predicted. This is also the case for other non-cancer illness: diagnosis (and likely treatment) at an older age is associated with younger. In contrast, the relationship between operations (be it first or second operation) and the predicted grey matter health goes in the opposite direction: operations performed at younger age appear to relate to a poorer predicted brain health.

For the remaining top contributing variables different patterns can be observed. In particular, for variables related to nutrition diet, while high coffee intake is associated with poorer predicted brain health, low dried fruit and cereal intake leads also to poorer predicted brain health. Finally, low hip circumference and low bone density are both associated with poorer brain health.”

(page 8, line 31- page 9, line 2):

“Beyond the critical effect of body health, our multivariate model also revealed the contribution of nutrition and diet on grey matter health. In particular, coffee, dried fruit, and cereals intake appear in the top most contributing factors consistently across algorithms. A higher coffee intake leads to poorer predicted brain health. Although a neuroprotective role for coffee is often discussed in the literature, our results demonstrate that a daily consumption above the moderate range (around 2 cups/day, the mean of the sample, see summary statistics in Supplementary Table 8) leads to a poorer predicted brain health.”

(page 9, line 25-32):

“In our machine learning models, hip circumference also appears to contribute to the prediction of individual grey matter health. More concretely, lower (than the mean of 102 cm, see Supplementary table 10) hip circumference is associated with poorer individual brain health in the prediction model. However, additional analyses show that, when taken in isolation, hip circumference is not associated with grey matter health in the main subset (null correlation with BAG, see Supplementary Figure 6). This suggests that hip circumference plays a role as a moderator variable in the individual prediction of grey matter health. In other words, lower hip

circumference on its own does not result in poorer brain health, but in interaction with other factors, it can lead to poorer predicted brain health.”

(page 21, line 7-13):

“Figure 3. SHAP value for the common most contributing variables when using random forest in the main subset (n = 3706, 261 exposome variables). The x axis represents the impact on the prediction: zero could be considered as the default prediction, while a shift to the right reflects a higher predicted BAG (i.e., poorer predicted brain health) and a shift to the left represents a lower predicted BAG (i.e., a better predicted grey matter health). Blue indicates a lower value for the variable, while pink/red indicates a higher value. The right panel summarizes which direction of the variable leads to poorer brain health (i.e. to higher BAG).”

[10] Sex/gender: Did the authors use self-reported sex or genetically inferred sex in their analyses? Unless a specific gender variable was used, referring to "sex" would be more appropriate in my view. Were discrepancies between self-reported and genetic sex taken into account in the analysis (e.g., by excluding individuals with reported vs. genetic sex-mismatch)?

In this study, we used self-reported sex (Data-Field 31) in line with previous related literature using UK Biobank^{15,18-20}. Because this self-related information was not cross-validated with genetic sex information (Data-Field 2200), we assumed that some participants may have reported their gender identity and therefore we used “sex/gender”. However, following the comment of the reviewer, we realized that the use of “sex/gender” can be misleading and we therefore now use “sex” terminology as suggested by the reviewer. Of note, we do not have access to Data-Field 2200, so we could not investigate potential discrepancies between self-reported and genetic sex.

(page 7, line 6-11):

“Finally, these patterns do not importantly influence by sex. Although in the main analysis, we treated sex as a covariate, in a supplementary analysis, we instead included it as a predictor variable in the model. We observed overall similar results. Furthermore, the contribution (SHAP value) of sex was very low, indicating minimal influence on the predictions. This suggests that the model’s performance remains stable and generalizable, regardless of whether sex is accounted for as a covariate or included as a predictor variable.”

[11] Interpretation of "almost null gap": The manuscript states that “most participants showed an almost null gap...,” which may suggest that individual brain-age gaps are negligible. However, the reported mean absolute error is approximately 3.9 years. Do the authors instead refer to the group-averaged mean of the BAG distribution, which may be close to zero by design? I suggest clarifying whether this statement refers to the distribution mean or to individual-level deviations and consider revising the wording to avoid potential confusion.

We thank the reviewer for pointing out the potential for misinterpretation regarding the 'almost null gap' statement. We have now revised the manuscript to clarify that this statement refers to the group-

averaged mean of the Brain Age Gap (BAG) distribution, which is close to zero by design (page 4, line 18-24):

“As could be expected this estimator was normally distributed (see Methods, Figure 5) with a group-averaged mean of zero. In the whole population, 90% of participants had BAG values between -9.20 and +10.18 years. Furthermore, 5% of participants had a BAG value inferior to -9.20 and 5% of participants had a BAG value superior to 10.18. This indicates that while the group-averaged mean BAG was close to zero, some participants showed a positive gap, indicating that their brain was estimated to be older than their chronological age, and some showed a negative gap, reflecting relatively preserved grey matter compared to their chronological age.”

[12] GitHub repository: I very much appreciate the inclusion of the GitHub repository, which appears well-organized, clear, and greatly enhances the transparency of the analysis workflow. The authors recommend using virtual environments such as conda, which is good practice. To further improve reproducibility, could the authors provide the corresponding environment.yml file(s) (or similar) with package versions? Additionally, it would be helpful to specify the system used to run the analyses.

We thank the reviewer for the positive feedback on our efforts to design and document the GitHub repository, as well as for the suggestion for completeness. The environment file has now been added to the GitHub repository.

Link:

https://github.com/MostafaMahdipour/Predicting_Brain_Age_Gap_BAG_using_UKB_exposome/blob/main/README.md

[13] The exposome concept: The concept of the exposome was relatively new to me, as I am more familiar with genome- and phenome-focused approaches. As I understand it, the exposome is generally defined as the totality of non-genetic internal and external exposures across the lifespan. However, many of the variables assessed in this study—such as dietary habits, smoking behaviour, or socioeconomic status—are known to have some degree of genetic influence. This suggests that the set of predictors used here may not reflect purely non-genetic exposures in the strictest sense. In my view, this overlap between genetic predisposition and environmental exposure highlights a broader conceptual limitation of the exposome framework. It might be helpful to briefly acknowledge this nuance in the discussion—perhaps in the context of causality, where the exposome framework alone does not resolve the challenge of causal inference, particularly in observational cohorts like the UK Biobank.

We agree with the reviewer that the exposome concept has some limitations. This is mainly an heuristic framework that promotes a holistic view on the range of factors that may influence a given aspect of human health. Accordingly, while in a phenome-wide approach (such as done in some previous UK Biobank studies)^{1,42} each variable is taken in isolation, the exposome concept insists on their co-occurrence, co-dependence, and potential interactions. This shift from reductionist to systemic perspectives aim to more realistically model the multifactorial nature of health and disease. However, we also agree with the reviewer that there is sometimes an intricate link between some exposome variables (in particular those pertaining to body health and to life style) and genetic factors, that prevents causal inferences. In line with reviewer’s comment, the limitation has now been noted in the revised manuscript:

(page 10, line 31- page 11, line 3):

“In the same vein, our study points to relatively new factors (coffee intake, dried fruits intake and hip circumference) that seem to interact with metabolic and cardiovascular risk factors in predicting brain structural health. However, although we initially considered body and life style factors from the lens of the exposome, many factors (such as metabolic and cardiovascular conditions) may obviously be importantly influenced by genetic factors. Future studies should thus crucially also include genetic factors, which have been found to also partly explain interindividual variability in brain health measured with BAG^{8,43,44}. Leveraging individual genetic profiles to examine to which extent the combination of the individual genotype and the individual exotype can predict individual brain health in aging with predictive modeling would hence be a crucial step towards precision brain health.”

Reviewer #1 (Remarks on code availability):

[14] I briefly reviewed the GitHub repository, and it appears well-organized, well-documented, and comprehensive. To further improve reproducibility, the addition of a conda environment file and basic system information would be beneficial.

Thank you again for the positive feedback and the useful suggestions.

Reviewer #2 (Remarks to the Author):

I thank the Editor and Authors for the chance to review this interesting paper! Please find my structured comments below.

Key results

This study presents an interesting approach to answering the question of “What predicts individual brain health?” by using rich data from the UK Biobank. Grey matter-based brain age models are trained and then used to predict brain age in a sub-sample of the UK Biobank. The difference between the individuals’ brain age and chronological age (brain age gap) is used as a proxy for brain health. In a further step, the brain age gap is predicted from a set of variables labelled here as the exposome. The authors present that exposome variable predict between 3%-5% of the variance in the brain age gaps. Using SHAP values, the authors present a ranking of the exposome features in explaining the brain age gap.

We thank the reviewer for the positive evaluation of our manuscript, as well as his/her comments which have contributed to improve its quality.

Validity

[1] While the manuscript draws on rich and high-quality data, there are multiple uncertainties around BAG still limiting its interpretability. The extent to which BAG reflects brain health remains unclear.

We thank the reviewer for raising this important point. We acknowledge the ongoing debate surrounding the interpretation of brain age gap and its validity as an indicator of individual brain health. In order to raise the reader’s awareness on the potential limitations of the interpretation of BAG, we have now discussed this point in the revised manuscript:

(page 11, line 4-19):

“Second, it should also be acknowledged that although BAG is a very convenient indicator of the degree of relative preservation or degeneration of a whole grey matter pattern, it remains a derived indicator instead of a direct measurement, which implies some noise. Furthermore, its conceptual validity and interpretability remain importantly discussed³⁵. First, a BAG framework assumes a unique aging trajectory and deviance (older brain age) from this trajectory is considered as advanced or pathological aging³⁵. Yet, this assumption can be questioned. Second and relatedly, some studies suggest that deviance (older brain age) may reflect brain phenotype present since early life instead of an aging phenotype³⁶. Overall, these considerations question the extent to which BAG can be used as an indicator of brain health in aging. With these conceptual limitations in mind, in this study, we did not use the popular “brain (biological) aging” concept when referring to BAG on purpose because it is not clear whether BAG represents any single phenomenon or set of phenomena that can be referred to as aging³⁵. Actually, more research on the underlying neurobiological properties detected by brain age prediction will be required to differentiate which phenomena BAG is in fact sensitive to. In that perspective, systematic comparison with other markers of neurocognitive health in future studies could contribute to

better understand the neurobiological phenomenon captured by each marker and how each and every marker is shaped by the exposome.”

[2] Additionally, the results indicate relatively weak associations between the exposome and BAG. These effect sizes were not discussed, which would however be important for context when presenting SHAP values and making conclusions about exposome data being able to predict brain health.

We agree with the reviewer that in our study the exposome explains a relatively small part of the variance in grey matter health (3 to 5%). However, it should be noted that the model performance that we report here are in out-of-sample prediction, this is of crucial relevance for interpreting our results in the context of the literature. To the best of our knowledge, out-of-sample model fit or prediction has never been shown in previous studies examining associations between some exposome factors and a marker of brain structure. Yet, predictive modelling with out-of-sample validation targeting other aspects of human phenotype, such as behavioral phenotype, typically show very limited prediction accuracy and variance explained⁴⁵. Thus, the out-of-sample effect size in this study is well in the range that could be expected for a predictive modelling approach with proper out-of-sample evaluation.

Furthermore, if we look at studies that have examined associations between exposome factors and markers of brain structure, when effect sizes are provided, these are typically very small. For example, in Smith et al. (2019)¹, the highest within sample correlation (observed for bone mineral density) is -0.10. As another example, alcohol consumption generally explained less than one 1% of variance in brain structural estimate^{46,47} in other studies. Along the same line, standardized betas of biomedical and life style variables in regression model of brain structural estimates are generally below 0.1 (eg: ^{28,48,49}) while the model is evaluated in the sample in which it has been fitted. Thus, the current study, actually show that when a wide range of exposome variable are combined with multivariate modeling, a greater part of variance in brain structural health can be explained.

In that context, despite the limited variance explained, we believe that the SHAP values still provide valuable insights into the relative importance of different exposures and can help generate hypotheses for future research as suggested in our discussion. Identifying even small contributions of modifiable factors could be valuable for informing targeted interventions.

Nevertheless, we acknowledge that, in the absolute, our ability to predict individual brain health/explain interindividual variability in brain health from the individual exptotype remain limited. Accordingly, this is now emphasize in our revised manuscript:

(page 10, line 25-29):

“Second, while the prediction accuracy in the out-of-sample test set is significant and demonstrates the model generalizability, the effect size (r value) remains relatively moderate. Relatedly, the explained variance in brain structural health (i.e. in BAG) in the test set is relatively limited (~5%). This suggests that additional factors should be taken into account.”

(page 10, line 35- page 11, line 3):

“Future studies should thus crucially also include genetic factors, which have been found to also partly explain interindividual variability in brain health measured with BAG^{8,43,44}. Leveraging individual genetic profiles to examine to which extent the combination of the individual genotype and the individual exptotype can predict individual brain health in aging with predictive modeling would hence be a crucial step towards precision brain health.”

Significance

[3] The integration of exposome data with neuroimaging-based brain age prediction is timely and potentially impactful, particularly in the context of identifying modifiable lifestyle or environmental factors associated with brain health. The use of large-scale population data (UK Biobank) and open methods further enhance the manuscript's relevance. However, the modest variance explained and the unresolved issues around the interpretation of BAG somewhat limit the strength of the conclusions. While the work is incremental in its current form, stronger positioning within the existing literature and a more critical discussion of the results would increase its contribution to the field. Nevertheless, the findings support a forming body of literature highlighting that the state of the brain (or potentially brain health) is multivariate, being a useful contribution to the field.

We thank the reviewer for their balanced assessment of our work. We appreciate the recognition of the timeliness and potential impact of our study, as well as the strengths of our approach. We also acknowledge the reviewer's concerns regarding the modest variance explained and the limitations of BAG. As detailed in our answer to comment [1] of first reviewer and below, we have now acknowledged these limitations in the revised manuscript.

*Following reviewer #1's and reviewer #2's comments, in the revised manuscript, we have now stronger positioned our work within the existing literature. This implies a clearer justification of the novelty of our work in the introduction by using a machine learning's approach that account for interaction between exposome factors and their non-linear relationships with brain health in the introduction. To further demonstrate the new insight provided by such an approach, we have also emphasized new findings relative to previous studies that have used a phenome-wide approach (i.e. studies that have tested the association of each variable with BAG separately). These new findings mainly suggest diet (coffee intake, cereals intake, dried fruits intake) and body composition factors (hip circumference as a proxy for gluteofemoral fat) that interact with the already well-established metabolic (diabetes) and cardiovascular risk factors factors, in either a detrimental (eg: coffee intake) or protective way (ie. dried fruits intake). **This additional discussion has been reported in our response to reviewer #1 first comment above, please see the answer to comment #1 of first reviewer.***

Data and methodology: The dataset (UK Biobank) is appropriate and well-regarded. However, important methodological details are missing or underspecified. These include:
[4] Rationale for the choice of machine learning algorithms.

We thank the reviewer for raising this point. When selecting the algorithms, we took into account brain age prediction literature and complementary among algorithms. Thus we selected ridge regression, support vector regression (SVR), and random forest (RF) due to their established use and effectiveness with tabular data in the field of brain age prediction³⁹⁻⁴¹. Using these well-established algorithms, as their performances are comparable⁴¹, allow us to compare our brain age prediction results with other studies in the field. We also focused on those three algorithms for exposome-based prediction of brain health because they offer complementary strengths: ridge regression provides regularization to prevent overfitting, SVR is very popular and relatively more robust to outliers than ridge regression, and random forest as an ensemble method, has the ability to capture non-linearity and complex interaction among features, furthermore by combining multiple decision trees it usually improves prediction accuracy and robustness. This information has now been added in the revised manuscript:

(page 16, line 10-14):

“We selected these three algorithms because they offer complementary strengths: random forest as an ensemble method, has the ability to capture non-linearity and complex interaction among features. Furthermore, by combining multiple decision trees it usually improves prediction accuracy and robustness. While ridge regression provides regularization to prevent overfitting, SVR is very popular and relatively more robust to outliers than ridge regression.”

[5] Whether training and test subsets were strictly independent.

We thank the reviewer for raising this important point. When designing the study, we took great care that the samples for brain age prediction model and the sample for exposome-based prediction of BAG were strictly independent, within each sample we further ensure that training and test subsets were strictly independent. Concretely, following the methodology described in Cole et al. (2020), we first created a 'healthy' subset of participants for training our brain age prediction models. The remaining participants formed the 'rest of population' set, which was used for subsequent BAG calculation using the trained models. Furthermore, within the 'healthy' training set, we employed an out-of-sample validation set that was never used during the training process, ensuring independence and preventing data leakage. A similar procedure was followed for BAG prediction in the population subsets. This complex design to prevent data leakage is now illustrated in Figure 4/Supplemental Figure 7 and below:

Figure 4 (Supplementary Figure 7). Study design for models' development and test preventing data leakage between models, as well as between training and test sets. A) Brain Age Model development, B) Exposome-BAG-Model.

[6] The method and covariates used for age-bias correction.

We appreciate the reviewer's suggestion to elaborate on our age-bias correction method. Recognizing the proportional bias described in previous studies^{1,15}—whereby the difference between chronological age and predicted brain age tends to be negatively correlated with chronological age—we implemented a linear correction procedure. Specifically, we calculated a regression line with chronological age as the predictor and predicted brain age as the outcome, using only the training dataset. The resulting slope and intercept were then used to adjust the predicted brain age values in the testing dataset. This adjustment involved subtracting the intercept from each predicted brain age and then dividing by the

slope. Following this age-bias correction, we calculated the brain-age gap (BAG) by subtracting chronological age from the adjusted predicted brain age. We have added a detailed description of this procedure to the manuscript to improve clarity and reproducibility.

Additionally, to further ensure that the calculated BAG did not contain statistical signal pertaining to age, we also regress out age and square of age in the exposome-based prediction of BAG.

Age Bias correction in the methods (page 14, line 12-22):

“Importantly, several studies have brought attention to an age bias/age dependency in brain age prediction requiring a so-called age-bias correction³⁷. This was done here by regressing out the effects of age on the predicted age³⁸. Specifically, we calculated a regression line with chronological age as the predictor and predicted brain age as the outcome, using only the training dataset. The resulting slope and intercept were then used to adjust the predicted brain age values in the testing dataset. This adjustment involved subtracting the intercept from each predicted brain age and then dividing by the slope. In other words, the slope and intercept of the regression line between chronological age and the predicted age were estimated in the training set and used to adjust the predicted age in the test and the population samples. As expected, and illustrated in supplementary Table 11, this age-bias correction procedure substantially reduced the artefactual correlation between the subsequently computed BAG and age.”

Age and sex as confounders in exposome-based predictive model of BAG (page 16, line 21-26):

“Controlling for covariates was performed on the exposome variables within the cross-validation framework (using estimated from the train set). The set of covariates/ confounds include age, square of age, sex, height, and volumetric scaling from T1 head image to standard space. Thus, all care was taken to neutralize the influence of usual confounds by using a normative indicator of grey matter health and by additionally controlling for covariates within the prediction analysis.”

[7] How exposome variables were selected and grouped (e.g., based on UKB codes or theoretical assumptions).

In response to the reviewer's comment, our selection of exposome variables involved a structured, multi-stage approach, building upon prior research and focusing on data quality. Initially, we conducted an extensive literature review, examining over 50 scientific articles to identify exposome variables linked to brain age gap (BAG) and brain health. This review guided our efforts to represent independent clusters of health-related domains. We then compiled a comprehensive list of available variables within the UK Biobank, mapping variables from other cohorts when possible. We applied preliminary exclusion criteria based on the UK Biobank Showcase to remove irrelevant categories and instances. From an initial pool of 659 Field IDs, we prioritized 397 covering key domains like anthropometry, diet, and lifestyle. Data extraction was performed using the ukbb_parser package, resulting in 2,599 columns due to multiple time points and categorical variable expansions. Data curation was then conducted, which included addressing missing data by retaining columns with at least 2,000 filled values (as informed by Missingno package analysis), verifying variable types, applying uniqueness and membership constraints, and excluding implausible responses. Finally, we defined three subsets for

analysis. This approach combined insights from the literature, data accessibility, and rigorous data curation to ensure a relevant and high-quality set of exposome variables.

For a more detailed, step-by-step explanation, please refer to our response to the comment #2 from Reviewer #1.

[8] Quality control steps for both exposome variables and MRI data.

1- MRI data: An internal data quality controlled is performed by the UK Biobank team before releasing Imaging-Derived-Phenotype. Importantly, UK Biobank only releases IDPs after internal QC. Although we did not used these IDPs in our study, all T1 images that were used have been selected from the participants for which IDPs have been released. Furthermore, from the calculation of grey matter volume of different parcels of the Brain using CAT 12⁵⁰ within an in-house developed framework for computationally reproducible processing of large-scale data (FAIRly big)⁵¹, some quality variables are available. These include IQR (Image Quality Rating), NCR (Noise-Contrast-Ratio), ICR (Inhomogeneity-Contrast-Ratio) and total GVM (Gray Matter Volume). Using these metrics we conducted a quality control (Figure 5). As illustrated below, the relationships between these metrics confirmed overall good data quality, with only a small number of participants showing values outside the main distribution (e.g., low NCR with high IQR, extreme ICR values, or unusually low GMV). In particular, the histogram of total GMV revealed a small tail on the lower end, which could suggest potential outliers. To identify whether these cases reflected segmentation errors or true biological variation, we visually inspected the corresponding T1 images. This confirmed that the scans were of sufficient technical quality, and the low GMV values reflected genuine brain atrophy patterns rather than artefacts. Thus, while our QC identified a small number of participants with atypical values, no participants were excluded solely on this basis, as the images themselves were of adequate quality for analysis.

Figure 5 Quality Control of T1 MRI data using outputs from CAT 12: (A) Relationship between Image Quality Rating (IQR) and Noise-Contrast-Ratio (NCR), (B) Scatterplot of IQR versus Inhomogeneity-Contrast-Ratio (ICR), and (C) Histogram of total grey matter (GM) volume.

2- Exposome variables:

We provided a detailed explanation of our quality control steps in our response to reviewer #1's comment. Briefly, data curation included: handling missing values, variable type (categorical vs. continuous) assignment, uniqueness constraint, aberrant value exclusion and uninformative value exclusion.

[9] Treatment of variable redundancy and imbalanced group sizes across exposome domains.

Following reviewer #1's suggestions, we performed hierarchical clustering and generated a correlation heatmap (see Figure 3/Supplementary Figure 8 and Figure 2) to assess the redundancy and relationships between the 261 exposome variables. The hierarchical clustering analysis (Figure 2) shows that the variables cluster into distinct groups, generally aligning with broad categories such as diet, body composition, socio-affective/mental health, and cardiovascular factors. The correlation heatmap (Figure 3/Supplementary Figure 8) reveals some clusters of correlated variables, but this is mainly for body morphology and composition variables. Overall, a substantial degree of heterogeneity can be observed, indicating that many variables exhibit relatively low correlations with one another.

We acknowledged that imbalance exists across exposome domains. For full transparency, this is now illustrated in a pie chart in Figure 6/Supplementary Figure 9 (see below). However, we did not perform

explicit redundancy reduction or weighting, because the Random Forest algorithm is inherently robust to multicollinearity and capable of selecting the most informative variables. Accordingly, our results show that there is no relationship between the extent to which a domain is represented (i.e. the number of variables from this domain) and the likelihood for these variables to appear as key predictors of brain health. For example, although socio-affective domain is overrepresented compared to cardiovascular health, variable form this latter was consistently picked up by the predictive model while variables from the former did not appear in the key findings.

Figure 6 (Supplementary Figure 9). Distribution of exposome variables across different exposome domains.

Following the reviewer’s comment, this information has been included in the revised manuscript (page 15, line 29- page 16, line 2):

“The redundancy between the 261 exposome variables were relatively low as illustrated by the correlation heatmap (Supplementary Figure 8). Although some clusters of correlated variables can be observed, these pertain to instrumentally related body morphology and composition variables and overall, a substantial degree of heterogeneity can be observed. The distribution of the variables across different exposome domains is illustrated for reader’s information in Supplementary Figure 9. Of note, from this set of variables, we did not perform any redundancy reduction since we subsequently used machine learning algorithms which are designed to be able to weight or select relevant features (see next section). “

[10] Additionally, it is unclear why brain age models were evaluated only with MAE and Pearson’s r, whereas exposome-to-BAG models used a broader set of metrics. A consistent evaluation framework would aid interpretability.

Regarding the use of different evaluation metrics for brain age models versus exposome-to-BAG models, we acknowledge the reviewer's point about consistency. However, for brain age prediction, a large body of literature exists, with the vast majority reporting MAE and Pearson's r. To facilitate comparison with these existing studies and avoid potential misunderstandings, we primarily reported these two scores. For the exposome-to-BAG models, as this is a new area with no established benchmarks, we reported a broader set of metrics to provide a more comprehensive evaluation and to potentially serve as a reference for future research.

[11] Again, while brain age provides a simplified metric of the state of the brain, which can be described as a proxy of brain health, its limitations need to be discussed in order to not mislead readers who are not familiar with the concept.

We thank the reviewer for raising this important point. We acknowledge the ongoing debate surrounding the interpretation of brain age gap and its validity as an indicator of individual brain health. In order to raise the reader's awareness on the potential limitations of the interpretation of BAG, we have now discussed this point in the revised manuscript:

(page 11, line 4-19):

“Second, it should also be acknowledged that although BAG is a very convenient indicator of the degree of relative preservation or degeneration of a whole grey matter pattern, it remains a derived indicator instead of a direct measurement, which implies some noise. Furthermore, its conceptual validity and interpretability remain importantly discussed³⁵. First, a BAG framework assumes a unique aging trajectory and deviance (older brain age) from this trajectory is considered as advanced or pathological aging³⁵. Yet, this assumption can be questioned. Second and relatedly, some studies suggest that deviance (older brain age) may reflect brain phenotype present since early life instead of an aging phenotype³⁶. Overall, these considerations question the extent to which BAG can be used as an indicator of brain health in aging. With these conceptual limitations in mind, in this study, we did not use the popular “brain (biological) aging” concept when referring to BAG on purpose because it is not clear whether BAG represents any single phenomenon or set of phenomena that can be referred to as aging³⁵. Actually, more research on the underlying neurobiological properties detected by brain age prediction will be required to differentiate which phenomena BAG is in fact sensitive to. In that perspective, systematic comparison with other markers of neurocognitive health in future studies could contribute to better understand the neurobiological phenomenon captured by each marker and how each and every marker is shaped by the exposome.”

Finally, I appreciate the authors' open science approach and well-organized GitHub repository!

Thanks a lot for the positive feedback on our GitHub repository.

Analytical approach

[12] The use of ensemble models such as Random Forests and SHAP value interpretation is appropriate. Nonetheless, the manuscript lacks information on the stability and robustness of the feature importances across methods. Are the SHAP results stable when compared to, for example, permutation importance or bootstrapped variable selection?

To address the reviewer’s comment and assess the stability of SHAP-derived feature importances, we performed a cross-validation–based stability analysis. Specifically, SHAP values were computed independently for the test set of each outer folds. For each fold, we calculated the mean absolute SHAP value per feature, resulting in fold-specific importance profiles.

We then quantified the similarity of these profiles across folds by computing pairwise correlations. The resulting fold–fold correlation matrix showed consistently high agreement, with an average correlation of 0.73 ± 0.10 and many fold–fold pairs exceeding 0.7 (Figure 7/Supplementary Figure 10). This demonstrates that the relative importance of features is stable across different resampling splits.

While we acknowledge that alternative methods (such as permutation importance or bootstrapped feature selection) could also be applied, our analysis confirms that SHAP importances are stable across sets. This stability analysis has now been included in the Supplemental material.

Figure 7 (Supplementary Figure 10). fold-to-fold mean absolute SHAP values’ correlations.

Following the reviewer's comment, this information has been included in the revised manuscript (page 16, line 35- page 17, line 2):

“Indeed, SHAP offers a cohesive framework for interpreting predictions in explainable Artificial Intelligence, assigning importance to each variable contributing to a prediction. The stability of SHAP patterns across folds is demonstrated in Supplementary Figure 10.”

[13] The choice of performance metrics should also be reconsidered. Reporting R^2 in the main text -- not only in the Supplement -- is essential for gauging practical significance. Moreover, the authors should ensure that statistical tests are labelled clearly; for example, the description of a t-test on a Pearson correlation is technically accurate but might be confusing in context.

Following the reviewer's comment, we have now included the R^2 values in the main manuscript (page 5, line 9-18):

“We could significantly predict individual grey matter health based on the exotype in the main subset ($r = 0.23$; $p = 0.002$; $r^2 = 0.05$; $p = 0.002$, FDR-corrected; Supplementary Table 5). We could replicate this achievement in the replication and variables-restricted subsets (which had less exposome variables but a higher number of participants) with similar accuracies (replication subset: $r = 0.23$; $p = 0.004$; $r^2 = 0.05$; $p = 0.004$; and variables-restricted subset: $r = 0.24$; $p = 0.002$; $r^2 = 0.06$; $p = 0.002$, FDR-corrected; Supplementary Table 5) confirming the robustness of our results across subsamples. Furthermore, individual grey matter health could also be predicted using alternative algorithms, although achieving numerically slightly lower prediction accuracy (with $r = 0.17$; $p = 0.002$; $r^2 = 0.03$; $p = 0.002$ for Ridge Regression and $r = 0.16$; $p = 0.002$; $r^2 = 0.02$; $p = 0.002$ for SVR; FDR-corrected; Supplementary Table 5).”

Additionally, as we agree with the reviewer that mentioning a t-test for a correlation value might be confusing without reference to the methods, we have now removed the notion of t-test from the results.

Suggested improvements

[14] Most importantly, the conceptual framing of brain age gap (BAG) could benefit from a more cautious and precise treatment. Currently, several statements risk implying that BAG reflects a direct measure of biological ageing, rather than deviation from a normative reference model. It would be helpful to more clearly separate BAG from ageing processes per se, especially in light of recent evidence questioning this link.

We appreciate the reviewer's suggestion for a more cautious framing of the brain age gap (BAG) 1. As we discussed in previous comments, we have now explicitly acknowledged the issues surrounding the interpretation of BAG as an indicator of biological aging in the revised manuscript (page 11, line 9-12):

“Second and relatedly, some studies suggest that deviance (older brain age) may reflect brain phenotype present since early life instead of an aging phenotype³⁶. Overall, these considerations question the extent to which BAG can be used as an indicator of brain health in aging.”

Furthermore, we have also revised our statements to avoid reference to a biological aging process in the revised manuscript.

(page 7, line 32-34):

In that context, our study more specifically suggests that alcohol consumption, smoking, high blood pressure and diabetes are the strongest risk factors for altered grey matter.

[15] In line with this, we suggest revising Figure 1 to avoid labeling the x-axis as “true BAG,” which may unintentionally suggest the existence of an objective ground truth.

We appreciate reviewer’s comment and to avoid misunderstanding we have now changed the label from “true BAG” to the “Target BAG”.

[16] The methodological transparency would also benefit from further detail. The rationale for selecting specific machine learning algorithms should be articulated.

As we discussed in our responses to Reviewer #1’s comment #7 and Reviewer #2’s comment #4, when selecting the algorithms, we took into account brain age prediction literature and complementary among algorithms. Thus we selected ridge regression, support vector regression (SVR), and random forest (RF) due to their established use and effectiveness with tabular data in the field of brain age prediction³⁹⁻⁴¹. Using these well-established algorithms, as their performances are comparable⁴¹, allow us to compare our brain age prediction results with other studies in the field. We also focused on those three algorithms for exposome-based prediction of brain health because they offer complementary strengths: ridge regression provides regularization to prevent overfitting, SVR is very popular and relatively more robust to outliers than ridge regression, and random forest as an ensemble method, has the ability to capture non-linearity and complex interaction among features, furthermore by combining multiple decision trees it usually improves prediction accuracy and robustness. This information has now been added in the revised manuscript as suggested by the reviewer:

(page 16, line 10-14):

“We selected these three algorithms because they offer complementary strengths: random forest as an ensemble method, has the ability to capture non-linearity and complex interaction among features. Furthermore, by combining multiple decision trees it usually improves prediction accuracy and robustness. While ridge regression provides regularization to prevent overfitting, SVR is very popular and relatively more robust to outliers than ridge regression.”

[17] along with an assessment of model stability and generalizability.

Following the reviewer’s comment, the stability of the SHAP values have now been demonstrated in Figure 7/Supplementary Figure 10.; Following the reviewer’s comment, this information has been included in the revised manuscript (page 16, line 35- page 17, line 2):

“Indeed, SHAP offers a cohesive framework for interpreting predictions in explainable Artificial Intelligence, assigning importance to each variable contributing to a prediction. The stability of SHAP patterns across folds is demonstrated in Supplementary Figure 10.”

Regarding the generalizability, the out-of-sample performance is a direct measure of model generalizability. This has now been clarified in the revised manuscript (page 10, line 25-27):

“Second, while the prediction accuracy in the out-of-sample test set is significant and demonstrates the model generalizability, the effect size (r value) remains relatively moderate.”

[19] Similarly, please specify how the age-bias correction was implemented—was it linear or non-linear, which covariates were included, and was the slope from the training set used? These details are important for reproducibility and interpretability.

Following the reviewer’s comment, we have now provided detailed explanation of how the age-bias correction was implemented (page 14, line 12-22):

“Importantly, several studies have brought attention to an age bias/age dependency in brain age prediction requiring a so-called age-bias correction³⁷. This was done here by regressing out the effects of age on the predicted age³⁸. Specifically, we calculated a regression line with chronological age as the predictor and predicted brain age as the outcome, using only the training dataset. The resulting slope and intercept were then used to adjust the predicted brain age values in the testing dataset. This adjustment involved subtracting the intercept from each predicted brain age and then dividing by the slope. In other words, the slope and intercept of the regression line between chronological age and the predicted age were estimated in the training set and used to adjust the predicted age in the test and the population samples. As expected, and illustrated in supplementary Table 11, this age-bias correction procedure substantially reduced the artefactual correlation between the subsequently computed BAG and age. “

[20] I also recommend expanding the description of the exposome variables, particularly regarding how they were selected, grouped, and whether any steps were taken to mitigate potential imbalance across groups (e.g., lifestyle vs. environmental exposures).

We appreciate the reviewer's suggestion to expand the description of the exposome variables and the selection process. As detailed in our response to Reviewer #1's comment #2 and further elaborated in our responses to Reviewer #2's comments #7 and #9, the selection of exposome variables was a multi-stage process, primarily driven by a desire to represent independent clusters of health-related domains based on existing literature.

The main steps (detailed above) are:

1. Literature Review
2. Initial Variable Listing
3. Preliminary Exclusion Criteria
4. Variable Prioritization
5. Data Extraction (parsing UK Biobank data)
6. Data Curation
7. Subset Definition

To further clarify this process, we have now provided more detailed descriptions in the revised manuscript (page 15, line 6-29):

“Exposome variables

To represent the exposome, we used a wide range of non-imaging variables encompassing biomedical, lifestyle, socio-economical and early life factors as illustrated in Figure 1 and the full list of exposome variables are available in Supplementary Table 2. The selection of variables was guided by previous publications reporting either associations (with correlation or regression usually) with brain health as measured by the brain age gap^{5,6,15,22-27} or associations with grey matter structure^{18,28-32} in the UK Biobank. This choice was also constrained by data availability. To avoid potential biases associated with substantial missing values across participants, we focused on variables that were available in at least 2000 participants including both males and females. We also performed some additional data curation processes, which included: uniqueness constraint (we ensured that each field ID and participant was represented uniquely in the dataset and we removed potential duplicate records), aberrant value exclusion (we verified that all variable values fell within the valid range or allowed set of categories e.g., no out-of-range numerical values or invalid category codes), and uninformative value exclusion (we excluded implausible or non-informative responses such as “do not know” or “prefer not to answer”). Furthermore, we also calculated duration variables for medical conditions by subtracting onset age from the age when attended assessment center (chronological age).

As illustrated in Figure 1, these procedures led us to define a main subset in which 261 distinct exposome variables were available for 3706 participants. By dropping out two variables (left and right heel bone density), we could create a bigger subset of 4292 participants that served as a replication subset. Finally, we also identified a bigger subset of 7736 participants with data for 201 distinct exposome variables. In this “variables-restricted subset”, mainly variables related to socio-affective and mental health domains were missing compared to the main and replication subsets. Standard preprocessing was performed on exposome variables including variables standardization (within the cross-validation scheme to avoid any data leakage). “

[21] Relatedly, it would be helpful to clarify whether weighting or redundancy reduction was applied to address overrepresentation or multicollinearity.

We have now clarified in the revised manuscript that no a-priori weighting or redundancy reduction has been applied:

(page 15, line 34- page 16, line 2):

“Of note, from this set of variables, we did not perform any redundancy reduction since we subsequently used machine learning algorithms which are designed to be able to weight or select relevant features (see next section).”

[22] While performance metrics like Pearson's r are useful, including R^2 values in the main text, rather than relegating them to the supplement, would give readers a more complete sense of the model's explanatory power.

Following the reviewer's comment, we have now incorporate the R^2 values alongside the Pearson's r values in the main manuscript (page 5, line 9-18):

“We could significantly predict individual grey matter health based on the exptotype in the main subset ($r = 0.23$; $p = 0.002$; $r^2 = 0.05$; $p = 0.002$, FDR-corrected; Supplementary Table 5). We could replicate this achievement in the replication and variables-restricted subsets (which had less exposome variables but a higher number of participants) with similar accuracies (replication subset: $r = 0.23$; $p = 0.004$; $r^2 = 0.05$; $p = 0.004$; and variables-restricted subset: $r = 0.24$; $p = 0.002$; $r^2 = 0.06$; $p = 0.002$, FDR-corrected; Supplementary Table 5) confirming the robustness of our results across subsamples. Furthermore, individual grey matter health could also be predicted using alternative algorithms, although achieving numerically slightly lower prediction accuracy (with $r = 0.17$; $p = 0.002$; $r^2 = 0.03$; $p = 0.002$ for Ridge Regression and $r = 0.16$; $p = 0.002$; $r^2 = 0.02$; $p = 0.002$ for SVR; FDR-corrected; Supplementary Table 5).”

[23] This is particularly relevant given that only a modest proportion of the variance in BAG is explained. These small effects should be clearly stated and considered when discussing the results.

Following the reviewer's suggestion, effect size have now been critically discussed in the revised manuscript (page 10, line 25-28):

“while the prediction accuracy in the out-of-sample test set is significant and demonstrates the model generalizability, the effect size (r value) remains relatively moderate. Relatedly, the explained variance in brain structural health (i.e. in BAG) in the test set is relatively limited (~5%).”

[24] Moreover, the limitations of BAG as a derived and relatively noisy variable should be more directly acknowledged and discussed in a dedicated limitations section.

Following all reviewers' comments, a specific section has now been devoted to point out the limitations of BAG (page 11, line 4-12):

“Second, it should also be acknowledged that although BAG is a very convenient indicator of the degree of relative preservation or degeneration of a whole grey matter pattern, it remains a derived indicator instead of a direct measurement, which implies some noise. Furthermore, its conceptual validity and interpretability remain importantly discussed⁴⁸. First, a BAG framework assumes a unique aging trajectory and deviance (older brain age) from this trajectory is considered as advanced or pathological aging⁴⁸. Yet, this assumption can be questioned. Second and relatedly, some studies suggest that deviance (older brain age) may reflect brain phenotype

present since early life instead of an aging phenotype⁴⁹. Overall, these considerations question the extent to which BAG can be used as an indicator of brain health in aging.”

[25] Finally, some sections of the Discussion could be condensed to reduce repetition and instead used to place the findings in context of comparable studies. Phrasing such as “optimal lifestyle” and broad claims like “all factors...constitute the exposome” could also be softened or made more precise to avoid overinterpretation. Addressing these points would, in my view, substantially enhance the clarity and impact of the manuscript.

As discussed in previous comments, we have now better situated our results in the context of previous literature. Following the reviewer’s comment, we have also carefully rephrased our statements in the discussion. For example, at page 3 line 18, we have replaced “optimal lifestyle” by “healthier lifestyle”. Furthermore, we have rephrased “all factors, collectively constitute the exposome” (page 2 line 32-34) as “while many of the individual factors mentioned above have long been studied in isolation, the exposome concept insists on their co-occurrence, co-dependence, and potential interactions”.

Clarity and context

[26] The manuscript is generally well-structured, but several areas require clarification. Some sentences need rephrasing for precision and scientific tone (e.g., “leads to a worst predicted grey matter health”).

*We have now replaced all instances of phrases like 'worst grey matter health' with more standard alternatives such as 'poorer brain health'. **Please see the answer to comment #9 of Reviewer #1.***

[27] The Discussion section currently repeats earlier results rather than synthesizing them with broader literature. A more critical and integrative discussion would improve clarity and impact.

Following the reviewers’ comments, we have now importantly revised the discussion. First, we have better integrated our findings with previous literature. Second, we have also further highlighted the novelty of our findings and their potential interpretation at the light of previous evidence of relationships between different biomedical factors. Third, we have also now discussed the potential limitations of our study, in particular when considering critical literature on the validity an interpretation of the brain age gap.

References

[28] Multiple references could be added to foster discussion in the Discussion section. This includes both brain age-specific and brain age-unspecific research.

In line with previous comments, we have now better acknowledged previous literature. In particular, we have reported previous phenome-wide associations studies in UK Biobank, we have also integrated other UK Biobank studies that have highlighted relationships between different biomedical factors and we have also included additional, more critical, pieces on the limitations of the brain age gap.

Reviewer #2 (Remarks on code availability):

[29] The GitHub repository is well organized and the most important bits of the code are available. I did not run the code, but after reading it, I do not see a reason why it would not run (at least after some refinements). Only critique: The code could be commented better, but this is more or less always the case for all code one is not writing oneself.

We thank the reviewer for the positive feedback on our GitHub repository. Following the reviewer's comment, we have further documented the code.

Reviewer #3 (Remarks to the Author):

In this paper, the authors investigated the association between exotypes and brain age gap. The use of the large UK biobank dataset is a key strength of the study. Further clarification and justification regarding the methodology, reporting and discussion of the results, particularly in the context of existing literature, is required.

We thank the reviewer for the positive evaluation of our manuscript, as well as his/her comments which have contributed to improve its quality.

[1] The authors suggest that to avoid potential biases associated with substantial missing values across participants, they focused on variables that were available in at least 2000 participants including both males and females - however, this would introduce bias, as has been well documented via complete cases analysis. Why was a robust method for addressing missing data not conducted, such as multiple imputation?

We appreciate the reviewer's concern about potential bias from restricting analyses to variables observed in $\geq 2,000$ participants. We agree that complete-case strategies can be biased. In this study, we considered multiple imputation (MI) carefully. However, we note several issues or challenges here for adopting this approach. First, as showed in Figure 1 (reported below), we encounter massive missing values for a wide range of variables.

Figure 1. Missingness pattern in the literature-based a-priori selected data fields (Missingno package). The gray colours are the filled values in women participants. Blue colours are the filled values in men participants. White colour reflects missing values, and red colour indicates non-informative answers (i.e., I don't know, I prefer not to answer etc.).

Indeed, as can be seen in Figure 1, the set of participants with neuroimaging data exhibits block-like patterns of missingness (large participant groups with many variables missing), which substantially limits the overlap of observed information across variables. In such settings, chained-equations MI is likely to become unstable, invalid or heavily model-dependent because many conditional models must be estimated with little shared data. Second, the imputation model should include all variables related to missingness and different variable types (continuous vs. categorical) typically require different conditional models (e.g., predictive mean matching, logistic/multinomial models) which would make the process extremely tedious and complex for a set of several hundreds of variables. This would also represent a major computational burden⁵². Finally, this massive preprocessing would have to be implemented without the cross-validation loop (i.e. by inferring parameters from the training to complete missing information in the training and test set) to avoid data leakage. Thus, such an approach

would substantially increase the complexity and computational demands of the analysis, while the benefit and remaining potential bias would remain uncertain. Considering these issues, we chose a minimum-availability filter ($\geq 2,000$ participants) to retain variables with sufficient observed support, reduce reliance on heavy modeling assumptions, and keep computation tractable.

[2] It is not clear to me how the statement “BAG does not rely on chronological age” is true, given the brain age gap subtracts the chronological age from the predicted age.

We thank the reviewer for pointing out that our statement can be misleading. We initially aimed to state that there was no information related to age in the Brain Age Gap, especially after age-bias correction. Following the reviewer’s comment, our statement has been rephrased to avoid misunderstanding in the revised manuscript (page 14, line 31-33):

“Moreover, it is cross-validated and does not convey age-related information (or age-related statistical signal) following age-bias correction, allowing it to directly assess deviations from population norms.”

[3] I do not see how an individualised approach has been adopted here. Fundamentally, these models show relationships between a set of variables and brain age gap, but do not provide an individualised approach to predicting brain age gap.

We appreciate the reviewer's concern regarding the degree to which our model provides individualized insights. We acknowledge that our primary analysis focuses on identifying generalizable relationships between exposome variables and brain age gap (BAG) at a population level. However, by their very nature, machine learning models are trained to infer an outcome in any new individual that it has not seen before. Thus, we can expect that in the future, if prediction accuracy increases, such models can be useful for the prediction of vulnerability and resilience at the individual level. The ability of a multivariate machine learning model to consider hundreds of interacting exposome variables in order to make predictions at the individual level can be seen from the examination of the SHAP values. To illustrate this, we have reported below (Figure 8) the SHAP values for the top 30 variables for two different participants who were both predicted to have higher BAG (i.e. poorer grey matter health). We can see from these plots that the contributing pattern for the first participant includes a long smoking duration, a long history of high blood pressure, a high coffee intake, cannabis at older age, asthma diagnosed at older age, pulmonary embolism diagnosis age, etc... while for the second participant, its contributing pattern includes a more recently diagnosed diabetes, a high pulse rate, a low water intake, a low cereals intake, etc... Such level of complexity to infer about individual brain outcome can not be captured by phenome-wide correlation approaches for example. However, we acknowledge that the current level of accuracy achieved does not provide fine prediction at the individual level for personalized recommendation/prevention/intervention at this stage and accordingly, we ensure that our statements about individual-level prediction in the revised manuscript pertain to the potential of exposome-based prediction in a future perspective and not to the results of the current study.

Figure 8 individualised patterns of how exposome variables affected person A and person B to have positive BAGs (poorer brain health) using waterfall plot.

[4] At what timepoint were the exposures measured, in addition to the imaging data? Was the time between measures considered in the model?

Different exposome variables may have been measured at different time points. We have included a column called 'Instance' in Supplementary Table 2. This column indicates the UK Biobank assessment visit from which the data for each variable was obtained.

The 'Instance' coding is based on the UK Biobank Showcase (instancing coding 2) and primarily reflects data collected during Assessment Centre Visits. The approximate time ranges for these instances are shown in Table 3.

Table 3 instancing coding 2 from UK Biobank showcase.

Index	Description
0	Initial assessment visit (2006-2010) at which participants were recruited and consent given
1	First repeat assessment visit (2012-13)
2	Imaging visit (2014+)
3	First repeat imaging visit (2019+)

In general, most of the exposome variables were measured at the initial assessment visit (2006-2010) or the first repeat assessment visit (2012-2013), while the imaging data was primarily collected during the imaging visit (2014+). For more information about instancing, please check this link: "What is an instance index?". In this study, we took care to select measurements that were taken before or at the same time of the imaging visit, but not later.

Regarding whether the time between measures was considered in the model: we did not explicitly model the time between exposure measurements and imaging data. However, we prioritized measurements that pertain to relatively chronic exposures when several types of measurement were available (for example we selected weekly intake information for diet over food intake over the 24h before the visit). Furthermore, to account for different time of exposure, we calculated duration (such as smoking

duration, diagnosis duration,...). By doing so we believe that the time between a given exposome measurement and the brain scan should not significantly influence the prediction. In other words, the small variations in the time between the exposome measurement across subjects is unlikely to influence the pattern of the results.

[5] The entire discussion is missing a meaningful discussion of effect sizes i.e., how many additional years in brain ageing were observed as a result of each additional year for the exposure? This is needed to provide a meaningful interpretation of the results.

We agree with the reviewer that effect size is a relevant issue to be discussed in this study. Following the reviewer's comment, we have now discussed the limited effect size of our model in out-of-sample test sets (page 10, line 25-29):

“while the prediction accuracy in the out-of-sample test set is significant and demonstrates the model generalizability, the effect size (r value) remains relatively moderate. Relatedly, the explained variance in brain structural health (i.e. in BAG) in the test set is relatively limited (~5%). This suggests that additional factors should be taken into account.”

Although our machine learning model allows us to derive an overall effect size for the exposome variables set in explaining brain structural health in test samples, it does not allow any inference at the exposome variable (i.e. feature) level. This is why we used a SHAP explainer. However, while the SHAP approach can inform us on which features matter most, and whether they generally increase or decrease predictions, it cannot indicate how much the outcome changes per unit increase in a feature in the classical regression sense.

[6] It is important that all statements are supported by citations/evidence

We thank the reviewer for raising out that point. We have now carefully checked that all statements are supported by citations.

• [7] Please include a descriptive table for variables used in study

Following the reviewer's request, two separate descriptive tables (one for continuous and one for categorical variables) have been included in the Supplemental material (for the main subset) in docx format (`Supplementary Table 3.docx` for continuous variables and `Supplementary Table 4.docx` for categorical variables).

Reviewer #3 (Remarks on code availability):

Code is well documented

Thank you for your positive feedback on our codes.

References

1. Smith, S.M., Vidaurre, D., Alfaro-Almagro, F., Nichols, T.E. & Miller, K.L. Estimation of brain age delta from brain imaging. *NeuroImage* **200**, 528-539 (2019).
2. Jawinski, P., *et al.* Genome-wide analysis of brain age identifies 25 associated loci and unveils relationships with mental and physical health. *medRxiv*, 2023.2012. 2026.23300533 (2023).
3. De Lange, A.-M.G., *et al.* Multimodal brain-age prediction and cardiovascular risk: The Whitehall II MRI sub-study. *NeuroImage* **222**, 117292 (2020).
4. Wagen, A.Z., *et al.* Life course, genetic, and neuropathological associations with brain age in the 1946 British Birth Cohort: a population-based study. *The Lancet Healthy Longevity* **3**, e607-e616 (2022).
5. Tian, Y.E., *et al.* Heterogeneous aging across multiple organ systems and prediction of chronic disease and mortality. *Nature Medicine* **29**, 1221-1231 (2023).
6. Vidal-Pineiro, D., *et al.* Individual variations in 'brain age' relate to early-life factors more than to longitudinal brain change. *elife* **10**, e69995 (2021).
7. Dintica, C.S., *et al.* Long-term depressive symptoms and midlife brain age. *Journal of affective disorders* **320**, 436-441 (2023).
8. Jawinski, P., *et al.* Genome-wide analysis of brain age identifies 59 associated loci and unveils relationships with mental and physical health. *Nature Aging*, 1-18 (2025).
9. Farina, F.R., *et al.* Next generation brain health: transforming global research and public health to promote prevention of dementia and reduce its risk in young adult populations. *The Lancet Healthy Longevity* (2024).
10. Kim, J., *et al.* Intraocular Pressure, Glaucoma, and Dietary Caffeine Consumption: A Gene-Diet Interaction Study from the UK Biobank. *Ophthalmology* **128**, 866-876 (2021).
11. Jiang, R., *et al.* Associations of physical frailty with health outcomes and brain structure in 483 033 middle-aged and older adults: a population-based study from the UK Biobank. *The Lancet Digital Health* (2023).
12. Agrawal, S., *et al.* BMI-adjusted adipose tissue volumes exhibit depot-specific and divergent associations with cardiometabolic diseases. *Nature Communications* **14**, 266 (2023).
13. Alser, M., Naja, K. & Elrayess, M.A. Mechanisms of body fat distribution and gluteal-femoral fat protection against metabolic disorders. *Frontiers in Nutrition* **11**, 1368966 (2024).
14. Zeng, Y., Cao, S. & Yang, H. Causal associations between dried fruit intake and cardiovascular disease: A Mendelian randomization study. *Front Cardiovasc Med* **10**, 1080252 (2023).
15. Cole, J.H. Multimodality neuroimaging brain-age in UK biobank: relationship to biomedical, lifestyle, and cognitive factors. *Neurobiology of aging* **92**, 34-42 (2020).
16. Cherbuin, N., *et al.* Optimal blood pressure keeps our brains younger. *Frontiers in Aging Neuroscience* **13**, 694982 (2021).
17. Gray, J.C., *et al.* Associations of cigarette smoking with gray and white matter in the UK Biobank. *Neuropsychopharmacology* **45**, 1215-1222 (2020).
18. Miller, K.L., *et al.* Multimodal population brain imaging in the UK Biobank prospective epidemiological study. *Nature neuroscience* **19**, 1523-1536 (2016).
19. Smith, S.M., *et al.* Brain aging comprises many modes of structural and functional change with distinct genetic and biophysical associations. *Elife* **9**, e52677 (2020).
20. Dinsdale, N.K., *et al.* Learning patterns of the ageing brain in MRI using deep convolutional networks. *NeuroImage* **224**, 117401 (2021).
21. Brandes, N., Linial, N. & Linial, M. PWAS: proteome-wide association study—linking genes and phenotypes by functional variation in proteins. *Genome biology* **21**, 173 (2020).
22. Dove, A., *et al.* Diabetes, Prediabetes, and Brain Aging: The Role of Healthy Lifestyle. *Diabetes Care* **47**, 1794-1802 (2024).
23. Dinsdale, N.K., *et al.* Learning patterns of the ageing brain in MRI using deep convolutional networks. *NeuroImage* **224**, 117401 (2021).
24. Kolbeinsson, A., *et al.* Accelerated MRI-predicted brain ageing and its associations with cardiometabolic and brain disorders. *Scientific Reports* **10**, 19940 (2020).

25. Ning, K., Zhao, L., Matloff, W., Sun, F. & Toga, A.W. Association of relative brain age with tobacco smoking, alcohol consumption, and genetic variants. *Scientific Reports* **10**, 10 (2020).
26. Stolicyn, A., *et al.* Comprehensive assessment of sleep duration, insomnia, and brain structure within the UK Biobank cohort. *Sleep* **47**, zsad274 (2024).
27. Mcavoy, E., Stanley, E.A., Winder, A.J., Wilms, M. & Forkert, N.D. Brain Aging in Patients With Cardiovascular Disease From the UK Biobank. *Human Brain Mapping* **46**, e70252 (2025).
28. Cox, S.R., *et al.* Associations between vascular risk factors and brain MRI indices in UK Biobank. *European heart journal* **40**, 2290-2300 (2019).
29. Daviet, R., *et al.* Associations between alcohol consumption and gray and white matter volumes in the UK Biobank. *Nature communications* **13**, 1175 (2022).
30. Ye, J., *et al.* Evaluating the effect of birth weight on brain volumes and depression: An observational and genetic study using UK Biobank cohort. *European Psychiatry* **63**, e73 (2020).
31. Tian, Y.E., Cole, J.H., Bullmore, E.T. & Zalesky, A. Brain, lifestyle and environmental pathways linking physical and mental health. *Nature Mental Health* **2**, 1250-1261 (2024).
32. Newby, D., *et al.* The relationship between isolated hypertension with brain volumes in UK Biobank. *Brain and behavior* **12**, e2525 (2022).
33. Peng, H., Gong, W., Beckmann, C.F., Vedaldi, A. & Smith, S.M. Accurate brain age prediction with lightweight deep neural networks. *Medical image analysis* **68**, 101871 (2021).
34. Schulz, M.-A., Siegel, N.T. & Ritter, K. Beyond Accuracy: Refining Brain-Age Models for Enhanced Disease Detection. *bioRxiv*, 2024.2003. 2028.587212 (2024).
35. Heinrichs, J.-H. Brain age prediction and the challenge of biological concepts of aging. *Neuroethics* **16**, 25 (2023).
36. Elliott, M.L., *et al.* Brain-age in midlife is associated with accelerated biological aging and cognitive decline in a longitudinal birth cohort. *Molecular psychiatry* **26**, 3829-3838 (2021).
37. Gaser, C., Kalc, P. & Cole, J.H. A perspective on brain-age estimation and its clinical promise. *Nat Comput Sci* **4**, 744-751 (2024).
38. de Lange, A.-M.G. & Cole, J.H. Commentary: Correction procedures in brain-age prediction. *NeuroImage: Clinical* **26**(2020).
39. Azzam, M., *et al.* A review of artificial intelligence-based brain age estimation and its applications for related diseases. *Briefings in Functional Genomics* **24**, elae042 (2025).
40. Da Costa, P.F., Dafflon, J. & Pinaya, W.H. Brain-age prediction using shallow machine learning: predictive analytics competition 2019. *Frontiers in psychiatry* **11**, 604478 (2020).
41. Kumari, L.S. & Sundarajan, R. A review on brain age prediction models. *Brain Research* **1823**, 148668 (2024).
42. Jawinski, P., *et al.* Genome-wide analysis of brain age identifies 59 associated loci and unveils relationships with mental and physical health. *Nature Aging* (2025).
43. Yi, F., *et al.* Genetically supported targets and drug repurposing for brain aging: A systematic study in the UK Biobank. *Science Advances* **11**, eadr3757 (2025).
44. Jónsson, B.A., *et al.* Brain age prediction using deep learning uncovers associated sequence variants. *Nature communications* **10**, 1-10 (2019).
45. Wu, J., Li, J., Eickhoff, S.B., Scheinost, D. & Genon, S. The challenges and prospects of brain-based prediction of behaviour. *Nature Human Behaviour* **7**, 1255–1264 (2023).
46. Daviet, R., *et al.* Associations between alcohol consumption and gray and white matter volumes in the UK Biobank. *Nature communications* **13**, 1-11 (2022).
47. Topiwala, A., Ebmeier, K.P., Maullin-Sapey, T. & Nichols, T.E. No safe level of alcohol consumption for brain health: observational cohort study of 25,378 UK Biobank participants. *medRxiv* (2021).
48. Hamer, M., Sharma, N. & Batty, G.D. Association of objectively measured physical activity with brain structure: UK Biobank study. *Journal of Internal Medicine* **284**, 439-443 (2018).
49. Harris, M.A., *et al.* Structural neuroimaging measures and lifetime depression across levels of phenotyping in UK biobank. *Translational psychiatry* **12**, 1-9 (2022).
50. Gaser, C., *et al.* CAT–A computational anatomy toolbox for the analysis of structural MRI data. *bioRxiv*, 2022.2006. 2011.495736 (2022).

51. Wagner, A.S., *et al.* FAIRly big: A framework for computationally reproducible processing of large-scale data. *Scientific data* **9**, 80 (2022).
52. Von Hippel, P.T. How many imputations do you need? A two-stage calculation using a quadratic rule. *Sociological Methods & Research* **49**, 699-718 (2020).

Dear Editors and Reviewers,

Thank you for your letter and for the reviewers' comments concerning our manuscript entitled "What predicts individual brain health? a machine learning study spanning the exposome". Your comments were helpful in raising important points that allowed us to finalize and further improve the manuscript. In the following pages, we provide our point-by-point responses to each of the reviewers' comments. Revisions in the text are shown using **Blue** colour.

Responses to the comments of Reviewers:

Reviewer #1:

Remarks to the Author

The authors responded to my revision requests with notable rigor and thoroughness, which substantially improved the manuscript. All my previous concerns have been fully addressed. I have only two remaining comments / tips for finalization.

Dear Dr. Jawinski, we thank you for your positive evaluation of our manuscript, as well as for your comments, which have contributed to improving its quality.

[1] Relatedness field: Unfortunately, the tabular data fields referring to relatedness, as mentioned by the authors, are not very informative. There is a file containing pre-calculated kinship coefficients that could previously be downloaded using the UK Biobank tool gfetch with the -rel flag. Within the Research Analysis Platform, this relatedness file including kinship coefficients can now be found under Bulk/Genotype Results/Genotype calls/ukb_rel.dat. Possibly something that could be considered in a future study. However, access to this file may require a Tier 3 plan.

We thank you for pointing out the availability of pre-calculated kinship coefficients in the UK Biobank. It is indeed a helpful suggestion. At the time of the present study, we did not have access to 'Tier 3 plan' with in our UK Biobank application (application ID: 41655). Therefore, we were unable to explicitly account for kinship coefficients in the current analyses. Following the reviewer's comment, we have now expanded the limitations section of the manuscript to clarify this point and to explicitly acknowledge that future studies could integrate these pre-calculated kinship coefficients to address familial relatedness more rigorously:

(page 10, lines 31-36):

"However, several limitations should be noted. First, at the methodological level, in this study we did not account for family relatedness. The UK Biobank includes several participants with first-, second-, and third-degree familial relationships which can potentially bias the results. The genetic relatedness between pairs of participants has been summarized in the pre-calculated kinship coefficients, which could be incorporated in future work to explicitly model or control for familial relatedness."

[2] P-values and permutations: Thank you for the clarification. With 500 permutations, the lowest achievable p-value is < 0.002 (i.e., when none of the permutations yields a similarly extreme test statistic). It may be worth mentioning this explicitly, as it explains why many p-values appear as 0.002. I also noticed that for SVR (linear) and r-squared, the corrected p-value is lower than the uncorrected p-value; please re-check this. Additionally, consider reporting all p-values in this table with three decimal places for consistency.

We thank you for highlighting the limitation imposed by the number of permutations on the achievable p-values. As correctly noted, when permutation testing is performed with 500 permutations, the smallest possible p-value that can be obtained is approximately 0.002. This explains why several results in Supplementary Table 5 show p-values at or close to this threshold.

Importantly, these p-values indicate that the observed prediction accuracies are significantly above chance level and therefore non-random. However, with the current number of permutations, it is not possible to further quantify how extremely non-random these effects are. Estimating more precise (i.e., smaller) p-values would require running a larger number of permutations.

We have now clarified this point explicitly in the manuscript, where we state that p-values were estimated using permutation testing with 500 permutations and that the minimum attainable p-value under this procedure is approximately 0.002.

(Page 5 lines 9-20):

“We could significantly predict grey matter health based on the exptype in the main subset (r = 0.23; p = 0.002; r2 = 0.05; p = 0.002, FDR-corrected; Supplementary Table 5). We could replicate this achievement in the replication and variables-restricted subsets (which had less exposome variables but a higher number of participants) with similar accuracies (replication subset: r = 0.23; p = 0.004 ; r2 = 0.05; p = 0.004; and variables-restricted subset: r = 0.24; p = 0.002; r2 = 0.06; p = 0.002, FDR-corrected; Supplementary Table 5) confirming the robustness of our results across subsamples. Furthermore, grey matter health could also be predicted using alternative algorithms, although achieving numerically slightly lower prediction accuracy (with r = 0.17; p = 0.002; r2 = 0.03; p = 0.002 for Ridge Regression and r = 0.16; p = 0.002; r2 = 0.02; p = 0.002 for SVR; FDR-corrected; Supplementary Table 5). It is noteworthy that p-values were estimated using permutation testing with 500 permutations. Consequently, the smallest p-value that can be obtained under this procedure is approximately 0.002, and the true p-values may be smaller if a larger number of permutations were used.”

Regarding the lower corrected p-value than the uncorrected p-value for SVR (linear) and r-squared, we confirmed that this was due to a typographical error (happening during files format conversion for the submission). We have now carefully re-checked all p-values and corrected the table accordingly. Finally, we have reported them with three decimal places for consistency (Table 1, supplementary table 5) as you suggested.

Table 1(supplementary table 5), Grey matter health prediction performance across different subsets/algorithms including main subset (n = 3706), replication subset (n = 4202), and variable-restricted subset (n = 7736).

Algorithm	Random Forest			SVR(Linear)	Ridge regression
	Main	Replication	Variables-restricted	Main	Main
Mean Absolute Error (MAE)	4.48	4.52	4.53	4.54	4.53
Uncorrected p-value	0.034	0.024	0.002	0.002	0.002
Corrected p-value	0.035	0.026	0.003	0.003	0.003
Mean Squared Error (MSE)	32.22	32.63	32.66	33.14	33.07
Uncorrected p-value	0.006	0.022	0.002	0.002	0.002
Corrected p-value	0.007	0.024	0.003	0.003	0.003
root Mean Squared Error (rMSE)	5.67	5.71	5.71	5.75	5.75
Uncorrected p-value	0.004	0.044	0.002	0.002	0.002
Corrected p-value	0.005	0.044	0.003	0.003	0.003
r ²	0.05	0.05	0.06	0.02	0.03
Uncorrected p-value	0.002	0.004	0.002	0.002	0.002

Corrected p-value	0.003	0.005	0.003	0.003	0.003
Pearson Correlation (r)	0.23	0.23	0.24	0.16	0.17
Uncorrected p-value	0.002	0.004	0.002	0.002	0.002
Corrected p-value	0.003	0.005	0.003	0.003	0.003

I congratulate the authors on this strong revision.

Best wishes,

Philippe Jawinski

We thank you again for your careful assessment of the revised manuscript and for your supportive and constructive final comments.

Remarks on code availability:

The GitHub repository is well-organized, well-documented, and comprehensive. As requested, the authors have added a conda environment file during the revision, which further enhances reproducibility.

Thank you for the positive assessment of the GitHub repository and code availability. We are glad that the added conda environment file improves the clarity and reproducibility of the analyses.

Reviewer #2:

Remarks to the Author:

The authors have addressed my comments satisfactorily. Congratulations on a well-executed study!
Thank you for the positive evaluation of our manuscript and thank you for your valuable comments.

Remarks on code availability:

The authors have addressed my comments satisfactorily. Congratulations on a well-executed study!
Thank you for your positive assessment of the code availability and reproducibility.

Reviewer #3:

Remarks to the Author:

The authors have made considerable progress in addressing the initial comments and should be commended for this. However, further clarification and justification is required for transparency.

[1] The authors' rationale for not conducting multiple imputation is understandable, but this justification must be included directly in the manuscript. This is important because the decision not to impute has implications for potential bias. A concise explanation of the missingness structure, the limitations of MI under these conditions, and the trade-offs involved should be made clear to readers.

We appreciate your concern about including a direct justification. Following your suggestion, we have now referred to the block-like pattern of the missingness in the data (included Supplementary Figure 9 in the supplement) and state the limitations of MI under these conditions, as well as the trade-off involved in our strategy in the revised manuscript. (page 16, lines 18-30):

“To avoid potential biases associated with substantial missing values across participants, we focused on variables that were available in at least 2000 participants including both males and females. Of note, we carefully considered multiple imputation. However, the missingness structure in the dataset posed substantial challenges. Specifically, participants with neuroimaging data exhibited block-like patterns of missingness, resulting in limited overlap across many variables (see Supplementary Figure 9). Under such conditions, chained-equations multiple imputation would be prone to instability and strong model dependence, as many conditional models would need to be estimated from sparse shared information. Moreover, implementing multiple imputation across several hundred heterogeneous variables (continuous and categorical) within a cross-validation framework—while avoiding data leakage—would substantially increase both methodological complexity and computational burden, with uncertain gains in bias reduction. Considering these limitations, we adopted a minimum-availability filtering strategy, prioritizing variables with sufficient observations while maintaining analytical tractability.”

Figure 1 (Supplementary Figure 9). Missingness pattern in the literature-based a-priori selected data fields (Missingno package). The grey colours are the filled values in women participants. Blue colours are the filled values in men participants. White colour reflects missing values, and red colour indicates non-informative answers (i.e., I don't know, I prefer not to answer etc.,)

[2] The response regarding the “individualised approach” is appreciated, but the manuscript should more clearly distinguish between what the current model can achieve and what may be possible in future applications throughout. At present, the predictive accuracy is not sufficient to support individual-level inference, and any statements implying personalised prediction should be reframed to refer explicitly to future potential rather than current capability.

We thank you for this important clarification. We agree that, given the current level of predictive accuracy, the model does not support individual-level inference or personalised prediction at present. We have therefore carefully revised the manuscript to clearly distinguish between the current capabilities of the model and the potential future applications of exposome-based predictive modeling for individual prediction.

Specifically, we now explicitly state that while the model generates individual-level predictions and explanations (e.g. SHAP values), these are intended to illustrate model behaviour rather than to support personalised clinical inference. All statements referring to individual-level prediction have been reframed to refer explicitly to future potential, contingent on substantially improved predictive accuracy.

Finally, we have also explicitly discussed the fact that our model currently does not support individual prediction for application (page 11, lines 13-21):

“Third, while the prediction accuracy in the out-of-sample test set is significant and demonstrates the model generalizability, the effect size (r value) remains relatively moderate. Relatedly, the explained variance in brain structural health (i.e. in BAG) in the test set is relatively limited (~5%). Although the model generates predictions and feature attributions at the individual level, the current level of predictive accuracy does not support personalized inference or individual-level decision-making. Accordingly, individual SHAP explanations are presented to illustrate model behaviour rather than to enable personalized interpretation. The potential utility of exposome-based machine learning models for individual-level vulnerability or resilience prediction should therefore be considered as a future perspective, contingent on substantial improvements in predictive performance.”

[3] The authors state that “small variations in the time between exposome measurement across subjects is unlikely to influence the pattern of results,” but this assumption needs to be transparently stated in the manuscript with the justification provided, as there is currently no evidence to support this.

We agree that the assumption regarding the timing of exposome measurements should be explicitly stated and justified in the manuscript.

In the present study, we did not explicitly model the time interval between exposome assessments and the imaging visit. Instead, when multiple measurements were available for a given variable, we prioritized indicators of relatively chronic exposure (e.g., weekly dietary intake rather than 24-hour recall), and where relevant we derived duration-based measures (e.g., smoking duration, disease duration). These choices were intended to reduce sensitivity to short-term fluctuations and partially account for between-subject variability in the timing of exposure measurement relative to brain imaging.

Nevertheless, this approach relies on the assumption that modest between-subject variation in the time interval between exposome assessment and imaging does not substantially alter the observed patterns. We acknowledge that this assumption cannot be directly tested within the current study and therefore represents a methodological limitation. Future longitudinal studies explicitly modeling time-varying exposures and exposure–outcome lag effects will be important to further clarify these temporal relationships.

(page 16 lines 31–35 and page 17 lines 1–5):

“We also performed some additional data curation processes, which included: uniqueness constraint (we ensured that each field ID and participant was represented uniquely in the dataset and we removed potential duplicate records), aberrant value exclusion (we verified that all variable values fell within the valid range or allowed set of categories e.g., no out-of-range numerical values or invalid category codes), and uninformative value exclusion (we excluded implausible or non-informative responses such as “do not know” or “prefer not to answer”). Furthermore, we also calculated duration variables for medical conditions by subtracting onset age from the age when attended assessment center (chronological age). Finally, it is noteworthy to mention that most exposome variables were measured at the initial UK Biobank assessment visit (2006–2010), and when data were not available at that time point, we used measurements from the first repeat assessment visit (2012–2013).”

(page 10 line 36 and page 11 lines 1-11):

“Second, as mentioned in method section, most exposome variables were measured at the initial UK Biobank assessment visit (2006–2010), and when data were not available at that time point, we used measurements from the first repeat assessment visit (2012–2013). We did not explicitly

model the time interval between exposome assessment and the imaging visit. To mitigate potential effects of temporal variability, we prioritized measures reflecting relatively chronic exposures (e.g., weekly dietary intake rather than 24-hour recall) and derived duration-based variables where possible (e.g., smoking duration, disease duration). Nevertheless, this approach assumes that modest between-subject variation in the timing of exposome assessment relative to imaging does not substantially influence the observed associations. This assumption cannot be formally tested within the present study and therefore represents a methodological limitation. Future longitudinal analyses explicitly modeling time-varying exposures and exposure–outcome lag effects will be important to further clarify these temporal relationships.”

[4] The discussion of effect sizes remains limited. While classical regression-style effect estimates cannot be derived from the modelling approach used, readers still require a meaningful way to interpret the magnitude and direction of key associations. The authors are encouraged to add brief clarifying text in the discussion noting these limitations, and include SHAP dependence plots or similar visualisations to illustrate how the most influential exposures relate to BAG across their value ranges.

We agree that the reviewer that the effect size may be of interest for some readers. It is noteworthy to mention that, although SHAP-based explanations help us in the interpretation of how changes in exposure values are associated with increases or decreases in predicted BAG across their observed ranges, but do not provide effect sizes in the traditional regression sense. We should here also mention that using SHAP dependence plots to infer about effect size can be misleading and leads to reader’s misunderstanding in our study for several reasons. First, typical SHAP dependence plots illustrate SHAP values for a given feature against the feature values (which does not provide information about associations between how exposure relates to BAG). Second, it is very important to note that confounds removal has been performed on the variables and therefore exposome variables values are actually adjusted standardized values. This leads for example to negative values for age at onset and duration which can be really difficult to interpret for the reader if SHAP dependence plots would be provided. Nevertheless to provide some information about effect size and the relationships between key exposome features and BAG, we used a traditional linear regression approach for our top 30 exposome features. The effect size of each top exposome variable on BAG (when accounting for confounds) is now illustrated by plotting standardized beta for each regression analysis together in Figure 1 (Supplementary Figure 7). It should be noted here that such model does not account for interaction between exposome features and non-linear relationships with BAG. Accordingly, the effect size of any single exposome variable on BAG can be considered as relatively small, in line with the literature on UK Biobank. This has also been discussed in the revised manuscript:

(Page 6, line 7–14):

“Additionally, to provide some information about effect size and the relationships between the 30 top exposome features and BAG, we perform a traditional linear regression approach. The effect size of each top exposome variable on BAG (when accounting for confounds) is illustrated by plotting standardized beta for each regression analysis in the supplementary figure 7. It should be noted here that such model does not account for interaction between exposome features and non-linear relationships with BAG. Accordingly, the effect size of any single exposome variable on BAG can be considered as relatively small, in line with the literature on UK Biobank. Whatever the approach, however, overall, most of the associations follow an expected pattern.”

(Page 11, lines 22–32):

“Along the same line, at the level of individual exposome variables, classical regression-style effect sizes (e.g. years of brain ageing per unit increase in exposure) cannot be directly inferred from the multivariate machine learning models used here, due to their non-linear structure and the complex interactions between predictors. To facilitate interpretation of the direction and shape of associations, we therefore relied on SHAP-based explanations, which illustrate how changes in exposure values are associated with increases or decreases in predicted BAG across their observed ranges, but do not provide effect sizes in the traditional regression sense. When considering each

of the top exposome variable individually in relation to BAG (supplementary figure 7) using a traditional linear regression model, the effect size of any single exposome variable on BAG can be considered as relatively small, in line with the literature on the UK Biobank. Overall, this reinforces the view that brain health emerges from the cumulative and interacting effects of many factors rather than large effects of single exposures.”

Figure 2 (supplementary figure 7). Standardized regression coefficients for the top 30 exposome features associated with brain age gap (BAG) in the main subset. The figure shows standardized beta coefficients derived from generalized linear models relating each exposome feature separately to BAG, when accounting for age, age², sex, height, and volumetric scaling from T1 head image to standard space. Both predictors and outcome were standardized to allow comparison of effect magnitudes across variables. Red dots illustrate significant associations after correcting for multiple comparisons (while blue dots are not significant). These analyses were conducted for reader’s information about the magnitude of each association when each exposome variable is considered in isolation in relation to BAG.

Remarks on code availability:

Have not re-reviewed code.

Thank you again for the positive feedback and the useful suggestions.